# Multi Time Scale World Models

**Vaisakh Shaj**[1][*]   **Saleh Gholam Zadeh**[1,2]   **Ozan Demir**[3]   **Luiz Ricardo Douat**[3]
**Gerhard Neumann**[1]
[1]Karlsruhe Institute Of Technology (KIT), Germany
[2]SAP SE, Germany
[3]Robert Bosch GmbH, Germany

## Abstract

Intelligent agents use internal world models to reason and make predictions about different courses of their actions at many scales [21]. Devising learning paradigms and architectures that allow machines to learn world models that operate at multiple levels of temporal abstractions while dealing with complex uncertainty predictions is a major technical hurdle [17]. In this work, we propose a probabilistic formalism to learn multi-time scale world models which we call the Multi Time Scale State Space (MTS3) model. Our model uses a computationally efficient inference scheme on multiple time scales for highly accurate long-horizon predictions and uncertainty estimates over several seconds into the future. Our experiments, which focus on action conditional long horizon future predictions, show that MTS3 outperforms recent methods on several system identification benchmarks including complex simulated and real-world dynamical systems. Code is available at this repository: https://github.com/ALRhub/MTS3.

## 1  Introduction

World models attempt to learn a compact and expressive representation of the environment dynamics from observed data. These models can predict possible future world states as a function of an imagined action sequence and are a key ingredient of model-predictive control [3] and model-based reinforcement learning (RL). One important dimension of world models is the level of temporal granularity or the time scale at which the model operates. Existing literature on world models operates at a single level of temporal abstraction, typically at a fine-grained level such as milliseconds. One drawback of single-time scale world models is that they may not capture longer-term trends and patterns in the data [17].

For efficient long-horizon prediction and planning, the model needs to predict at multiple levels of temporal abstractions [27, 22]. Intuitively, low-level temporal abstractions should contain precise details about the input so as to predict accurately in the short term, while high-level, abstract representations should simplify accurate long-term predictions. Both abstractions must also interrelate with each other at least in the sense that the higher-level predictions/plans can be turned into low-level moment-by-moment predictions. For example, in robotic manipulation, the robot must be able to perform precise and coordinated movements to grasp and manipulate the object at a fast time scale while at a slower time scale, the robot must also be able to recognize and utilize higher-level patterns and structures in the task, such as the shape, size and location of objects, and the overall goal of the manipulation task.

Furthermore, temporal abstractions can capture relevant task structures across dynamical systems under non-stationary which can be used to identify the similarities and differences between tasks, allowing the robot to transfer knowledge learned from one task to another [26, 17].

---

[*]Corresponding author. Email to <vaisakhs.shaj@gmail.com>.

37th Conference on Neural Information Processing Systems (NeurIPS 2023).

In this work, we attempt to come up with a principled probabilistic formalism for learning such multi-time scale world models as a hierarchical sequential latent variable model. We show that such models can better capture the complex, non-linear dynamics of a system more efficiently and robustly than models that learn on a single timescale. This is exemplified in several challenging simulated and real-world prediction tasks such as the D4RL dataset, a simulated mobile robot and real manipulators including data from heavy machinery excavators.

## 2 Preliminaries

State space models (SSMs) are Bayesian probabilistic graphical models [15, 12] that are popular for learning patterns and predicting behaviour in sequential data and dynamical systems. Formally, we define a state space model as a tuple $(\mathcal{Z}, \mathcal{A}, \mathcal{O}, f, h, \Delta t)$, where $\mathcal{Z}$ is the state space, $\mathcal{A}$ the action space and $\mathcal{O}$ the observation space of the SSM. The parameter $\Delta t$ denotes the discretization time-step and $f$ and $h$ the dynamics and observation models respectively. We will consider the Gaussian state space model that is represented using the following equations

$$\boldsymbol{z}_t = f(\boldsymbol{z}_{t-1}, \boldsymbol{a}_{t-1}) + \boldsymbol{\epsilon}_t, \quad \boldsymbol{\epsilon}_t \sim \mathcal{N}(\boldsymbol{0}, \boldsymbol{\Sigma_z}), \quad \text{and} \quad \boldsymbol{o}_t = h(\boldsymbol{z}_t) + \boldsymbol{v}_t, \quad \boldsymbol{v}_t \sim \mathcal{N}(\boldsymbol{0}, \boldsymbol{\Sigma_o}).$$

Here $\boldsymbol{z}_t \in \mathcal{Z}$, $\boldsymbol{a}_t \in \mathcal{A}$ and $\boldsymbol{o}_t \in \mathcal{O}$ are the latent states, actions and observations at time t. The vectors $\boldsymbol{\epsilon}_t$ and $\boldsymbol{v}_t$ denote zero-mean Gaussian noise. When $f$ and $h$ are linear/locally linear, inference can be performed efficiently via exact inference. There have been several works recently [9, 1, 25] where these closed-form solutions are coded as layers of a neural network in deep-state space model literature, i.e, the architecture of the network is informed by the structure of the probabilistic state estimator. Following this line of work, we propose a multi-time scale linear Gaussian state space model, whose inference can be performed via closed-form solutions.

**(Locally-)Linear Gaussian SSMs.** To perform inference in SSMs, we follow [1]. They use a (locally-)linear dynamics model. Moreover, they replace the observations $\boldsymbol{o}$ with a latent observation $\boldsymbol{w}$. This latent observation is obtained by an encoder $\boldsymbol{w}_{o_t} = \text{enc}_w(\boldsymbol{o}_t)$ along with the uncertainty of this observation, i.e., $\boldsymbol{\sigma}_{o_t} = \text{enc}_\sigma(\boldsymbol{o}_t)$. Due to the non-linear observation encoder, a simplified linear observation model can now be used. Hence, the dynamics and observation models can be described as

$$p(\boldsymbol{z}_{t+1}|\boldsymbol{z}_t, \boldsymbol{a}_t) = \mathcal{N}(\boldsymbol{A}_t \boldsymbol{z}_t + \boldsymbol{c}_t + \boldsymbol{B}_t \boldsymbol{a}_t, \text{diag}(\boldsymbol{\sigma}_z)), \text{ and } p(\boldsymbol{w}_{o_t}|\boldsymbol{z}_t) = \mathcal{N}(\boldsymbol{H}\boldsymbol{z}_t, \text{diag}(\boldsymbol{\sigma}_{o_t})),$$

where a simple observation matrix of $\boldsymbol{H} = [\boldsymbol{I}, \boldsymbol{0}]$ is used. The underlying assumption behind this observation model is that the latent state $\boldsymbol{z}_t = [\boldsymbol{p}_t^T, \boldsymbol{d}_t^T]^T$ has twice the dimensionality of the latent observation $\boldsymbol{w}_t$ and only the first half of the latent state, i.e., $\boldsymbol{p}_t$, can be observed. The second half of the latent state, i.e., $\boldsymbol{d}_t$, serves as derivative or velocity units that can be used by the model to estimate the change of the observable part of the latent state.

**Factorized Inference in Linear Gaussian SSMs.** Inference in the introduced linear Gaussian SSM is straightforward and can be performed using Kalman prediction and observation updates. However, these updates involve high dimensional matrix inversions that are expensive to evaluate and hard to backpropagate for end-to-end learning. Hence, [1] introduce a factorization of the belief $p(\boldsymbol{z}_t|\boldsymbol{o}_{1:t}, \boldsymbol{a}_{1:t-1}) = \mathcal{N}(\boldsymbol{\mu}_t, \boldsymbol{\Sigma}_t)$ such that only the diagonal and one off-diagonal vector of the covariance need to be computed, i.e.

$$\boldsymbol{\Sigma}_t = \left[ \begin{array}{cc} \boldsymbol{\Sigma}_t^u & \boldsymbol{\Sigma}_t^s \\ \boldsymbol{\Sigma}_t^s & \boldsymbol{\Sigma}_t^l \end{array} \right], \text{ with } \boldsymbol{\Sigma}_u = \text{diag}(\boldsymbol{\sigma}_t^s), \ \boldsymbol{\Sigma}_l = \text{diag}(\boldsymbol{\sigma}_t^l) \text{ and } \boldsymbol{\Sigma}_s = \text{diag}(\boldsymbol{\sigma}_t^s).$$

Using this factorization assumption, closed-form Gaussian inference can be performed using only scalar divisions which are fast and easy to back-propagate. These factorization assumptions form the basis for the inference update in our MTS3 model.

**Bayesian Aggregation.** To aggregate information from several observations into a consistent representation, [29] introduce Bayesian aggregation in the context of Meta-Learning. They again use an encoder to obtain a latent observation vector $\boldsymbol{r}_{o_t}$ and its uncertainty vector $\boldsymbol{\sigma}_{o_t}$. Given the observation model $p(\boldsymbol{r}_{o_t}|\boldsymbol{z}) = \mathcal{N}(\boldsymbol{H}\boldsymbol{z}, \text{diag}(\boldsymbol{\sigma}_{o_t}))$ with $\boldsymbol{H} = \boldsymbol{I}$ and a prior $p(\boldsymbol{z}) = \mathcal{N}(\boldsymbol{\mu}_0, \text{diag}(\boldsymbol{\sigma}_0))$, the posterior $p(\boldsymbol{z}|\boldsymbol{r}_{o_1:o_t})$ can again be effectively computed by Gaussian inference that involve only

scalar inversions. Note that computing this posterior is a simplified case of the Kalman update rule used in Gaussian SSMs [1], with no memory units, $\boldsymbol{H} = \boldsymbol{I}$ and no dynamics. To increase efficiency, the update rule can be formulated in a batch manner for parallel processing instead of an incremental update [29].

# 3   Multi Time Scale State Space Models

Our goal is to learn a principled sequential latent variable model that can model the dynamics of partially observable robotic systems under multiple levels of temporal abstractions. To do so, we introduce a new formalism, called Multi Time Scale State Space (MTS3) Model, with the following desiderata: i) It is capable of modelling dynamics at multiple time scales. ii) It allows for a single global model to be learned that can be shared across changing configurations of the environments. iii) It can give accurate long-term predictions and uncertainty estimates. iv) It is probabilistically principled yet scalable during learning and inference.

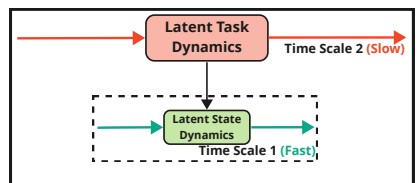

Figure 1: MTS3 captures slow-moving long-term trends as the latent task dynamics and the fast-moving short-term trends as the latent state dynamics.

## 3.1   General Definition

An MTS3 model with 2 timescales is defined by two SSMs on a fast and a slow time scale respectively. Both SSMs are coupled via the latent state of the slow time scale SSM, which parametrizes/"reconfigures" the system dynamics of the fast time scale SSM. While the fast time scale SSM runs at the original time step $\Delta t$ of the dynamical system, the slow time scale SSM is only updated every $H$ step, i.e., the slow time scale time step is given by $H\Delta t$. We will derive closed-form Gaussian inference for obtaining the beliefs for both time scales, resulting in variations of the Kalman update rule which are also fully differentiable and used to back-propagate the error signal [1, 9]. The definition with a 2-level MTS3 along with the inference and learning schemes that we propose is directly extendable to an arbitrary number of temporal abstractions by introducing additional feudal [5] hierarchies with longer discretization steps and is further detailed in Section 3.4.

### 3.1.1   Fast time-scale SSM

The fast time-scale (fts) SSM is given by $\mathcal{S}_{\text{fast}} = (\mathcal{Z}, \mathcal{A}, \mathcal{O}, f_{\boldsymbol{l}}^{\text{fts}}, h^{\text{fts}}, \Delta t, \mathcal{L})$. Here, $\boldsymbol{l} \in \mathcal{L}$ is a task descriptor that parametrizes the dynamics model of the SSM and is held constant for H steps. We will denote the task descriptor for the $k$th time window of $H$ steps as $\boldsymbol{l}_k$. The probabilistic dynamics and observation model of the fast time scale for the $t$th time step in the $k$th window can then be described as

$$p(\boldsymbol{z}_{k,t}|\boldsymbol{z}_{k,t-1}, \boldsymbol{a}_{k,t-1}, \boldsymbol{l}_k) = \mathcal{N}(f_{\boldsymbol{l}}^{\text{fts}}(\boldsymbol{z}_{k,t-1}, \boldsymbol{a}_{k,t-1}, \boldsymbol{l}_k), \mathbf{Q}), \text{ and}$$
$$p(\boldsymbol{o}_{k,t}|\boldsymbol{z}_{k,t}) = \mathcal{N}(h^{\text{fts}}(\boldsymbol{z}_{k,t}), \mathbf{R}). \tag{1}$$

**Task-conditioned marginal transition model.**   Moreover, we have to consider the uncertainty in the task descriptor (which will, in the end, be estimated by the slow time scale model), i.e., instead of considering a single task descriptor $\boldsymbol{l}_k$, we have to consider a distribution over task-descriptors $p(\boldsymbol{l}_k)$ for inference in the fts-SSM. This distribution will be provided by the slow-time scale SSM for every time window $k$. We can further define the marginal task-conditioned transition model for time window $k$ which is given by

$$p_{\boldsymbol{l}_k}(\boldsymbol{z}_{k,t}|\boldsymbol{z}_{k,t-1}, \boldsymbol{a}_{k,t-1}) = \int p(\boldsymbol{z}_{k,t}|\boldsymbol{z}_{k,t-1}, \boldsymbol{a}_{k,t-1}, \boldsymbol{l}_k)p(\boldsymbol{l}_k)d\boldsymbol{l}_k \tag{2}$$

**Latent observations.**   Following [1], we replace the observations by latent observations and their uncertainty, i.e., we use latent observation encoders to obtain $\boldsymbol{w}_{k,t} = \text{enc}_w(\boldsymbol{o}_{k,t})$ and an uncertainty encoder $\boldsymbol{\sigma}_{k,t} = \text{enc}_\sigma(\boldsymbol{o}_{k,t})$. The observation model is hence given by $p(\boldsymbol{w}_{k,t}|\boldsymbol{z}_{k,t}) = \mathcal{N}(h^{\text{fts}}(\boldsymbol{z}_{k,t}), \text{diag}(\boldsymbol{\sigma}_{k,t}))$.

### 3.1.2 Slow time-scale SSM

The slow time-scale (sts) SSM only updates every H time step and uses the task parameter $l$ as latent state representation. Formally, the SSM is defined as $\mathcal{S}_{\text{slow}} = (\mathcal{L}, \mathcal{E}, \mathcal{T}, f^{\text{sts}}, h^{\text{sts}}, H\Delta t)$. It uses an abstract observation $\beta \in \mathcal{B}$ and abstract action $\alpha \in \mathcal{A}$ that summarize the observations and actions respectively throughout the current time window. The general dynamics model is hence given by

$$p(l_k|l_{k-1}, \alpha_k) = \mathcal{N}(f^{\text{sts}}(l_{k-1}, \alpha_k), S). \tag{3}$$

While there exist many ways to implement the abstraction of observations and actions of the time windows, we choose to use a consistent formulation by fusing the information from all $H$ time steps of time window $k$ using Gaussian conditioning.

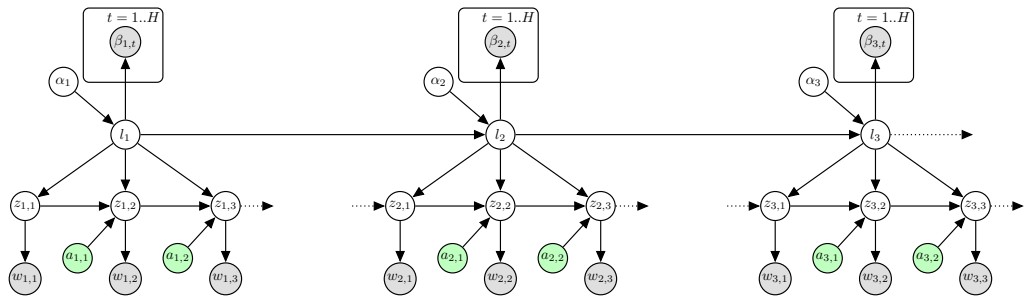

Figure 2: The graphical model corresponding to an MTS3 with 2 timescales. The latent task variable $l_k$ captures the slow-changing dynamics using abstract observation inferred from $\{\beta_{k,t}\}_{t=1}^{H}$ and abstract action $\alpha_k$ as described in section 3.1.2. The inference in the fast time scale uses primitive observations $w_{k,t}$, primitive actions $a_{k,t}$ and the latent task descriptor $l_k$ which parameterizes the fast-changing dynamics of $z_{k,t}$ for a time window k as discussed in the section 3.2.

**Observation abstraction.** In terms of the abstract observation model, we choose to model $H$ observations $\beta_{k,t}$, $t \in [1, H]$ for a single slow-scale time step $k$. All these observations can then be straightforwardly integrated into the belief state representation using incremental observation updates. The abstract observation and its uncertainty for time step $t$ is again obtained by an encoder architecture, i.e,

$$\beta_{k,t} = \text{enc}_\beta(o_{k,t}, t), \quad \nu_{k,t} = \text{enc}_\nu(o_{k,t}, t),$$

and $p(\beta_{k,t}|l_k) = \mathcal{N}(h^{\text{sts}}(l_k), \text{diag}(\nu_{k,t}))$. Hence, the abstract observation $\beta_{k,t}$ contains the actual observation $o_{k,t}$ at time step $t$ as well as a temporal encoding for the time-step. While multiple Bayesian observation updates are permutation invariant, the temporal encoding preserves the relative time information between the observations, similar to current transformer architectures.

**Action abstraction.** The abstract action $\alpha_k$ causes the transitions to the latent task $l_k$ from $l_{k-1}$. It should contain the relevant information of all primitive actions $a_{k,t}$, $t \in [1, H]$ executed in the time window $k$. To do so, we again use Bayesian conditioning and latent action encoding. Each control action $a_{k,t}$ and the encoding of time-step $t$ is encoded into its latent representation and its uncertainty estimate, i.e.,

$$\alpha_{k,t} = \text{enc}_\alpha(a_{k,t}, t), \quad \rho_{k,t} = \text{enc}_\rho(a_{k,t}, t).$$

The single latent actions $\alpha_{k,t}$ can be aggregated into a consistent representation $\alpha_k$ using Bayesian aggregation [29]. To do so, we use the likelihood $p(\alpha_{k,t}|\alpha_k) = \mathcal{N}(\alpha_k, \text{diag}(\rho_{k,t}))$ and obtain the posterior $p(\alpha_k|\alpha_{k,1:H}) = \mathcal{N}(\mu_{\alpha_k}, \Sigma_{\alpha_k})$, which is obtained by following the standard Bayesian aggregation equations, see Appendix A. Note that our abstract action representation also contains an uncertainty estimate which can be used to express different effects of the actions on the uncertainty of the prediction. Due to the Gaussian representations, we can compute the marginal transition model

$$p_{\alpha_k}(l_k|l_{k-1}, \alpha_{k,1:H}) = \int p_{\alpha_k}(l_k|l_{k-1}, \alpha_k)p(\alpha_k|\alpha_{k,1:H})d\alpha_k. \tag{4}$$

This transition model is used for inference and its parameters are learned.

### 3.1.3 Connecting both SSMs via inference

In the upcoming sections, we will devise Bayesian update rules to obtain the prior $p(l_k|\boldsymbol{\beta}_{1:k-1}, \boldsymbol{\alpha}_{1:k})$ and posterior $p(l_k|\boldsymbol{\beta}_{1:k}, \boldsymbol{\alpha}_{1:k})$ belief state for the sts-SSM as well as the belief states for the fts-SSM. The prior belief $p(l_k|\boldsymbol{\beta}_{1:k-1}, \boldsymbol{\alpha}_{1:k})$ contains all information up to time window $k-1$ and serves as a distribution over the task-descriptor of the fts-SSM, which connects both SSMs. This connection allows us to learn both SSMs jointly in an end-to-end manner.

The probabilistic graphical model of our MTS3 model is depicted in Figure 2. In the next section, we will present the detailed realization of each SSM to perform closed-form Gaussian inference and end-to-end learning on both time scales.

### 3.2 Inference in the Fast Time-Scale SSM

The fts-SSM performs inference for a given time window $k$ of horizon length $H$. To keep the notation uncluttered, we will also omit the time-window index $k$ whenever the context is clear. We use a linear Gaussian task conditional transition model, i.e,

$$p(\boldsymbol{z}_t|\boldsymbol{z}_{t-1}, \boldsymbol{a}_{t-1}, l_k) = \mathcal{N}\left(\mathbf{A}\boldsymbol{z}_{t-1} + \mathbf{B}\boldsymbol{a}_{t-1} + \mathbf{C}l_k, \mathbf{Q}\right), \tag{5}$$

where $\boldsymbol{A}$, $\boldsymbol{B}$, $\boldsymbol{C}$ and $\boldsymbol{Q}$ are state-independent but learnable parameters. In our formulation, the task descriptor can only linearly modify the dynamics which was sufficient to obtain state-of-the-art performance in our experiments, but more complex parametrizations, such as locally linear models, would also be feasible. Following [1], we split the latent state $\boldsymbol{z}_t = [\boldsymbol{p}_t, \boldsymbol{m}_t]^T$ into its observable part $\boldsymbol{p}_t$ and a part $\boldsymbol{m}_t$ that needs to be observed over time. We also use a linear observation model $p(\boldsymbol{w}_t|\boldsymbol{z}_t) = \mathcal{N}(\boldsymbol{H}\boldsymbol{z}_t, \mathrm{diag}(\boldsymbol{\sigma}_t))$ with $\boldsymbol{H} = [\boldsymbol{I}, \boldsymbol{0}]$.

We will assume that the distribution over the task descriptor is also given by a Gaussian distribution, i.e., $p(l_k) = \mathcal{N}(\boldsymbol{\mu}_{l_k}, \boldsymbol{\Sigma}_{l_k})$, which will be provided by the slow-time scale (sts) SSM, see Section 3.3. Given these modelling assumptions, the task variable can now be integrated out in closed form, resulting in the following task-conditioned marginal transition model

$$p_{l_k}(\boldsymbol{z}_t|\boldsymbol{z}_{t-1}, \boldsymbol{a}_{t-1}) = \mathcal{N}\left(\mathbf{A}\boldsymbol{z}_{t-1} + \mathbf{B}\boldsymbol{a}_{t-1} + \mathbf{C}\boldsymbol{\mu}_{l_k}, \mathbf{Q} + C\boldsymbol{\Sigma}_{l_k}C^T\right), \tag{6}$$

which will be used instead of the standard dynamics equations. We follow the same factorization assumptions as in [1] and only estimate the diagonal elements of the block matrices of the covariance matrix of the belief, see Appendix B. The update equations for the Kalman prediction and observation updates are therefore equivalent to the RKN [1].

### 3.3 Inference in the Slow-Time Scale SSM

**Prediction Update.** We follow the same Gaussian inference scheme as for the fts-SSM, i.e., we again employ a linear dynamics model $p(l_k|l_{k-1}, \boldsymbol{\alpha}_k) = \mathcal{N}(\boldsymbol{X}l_{k-1} + \boldsymbol{Y}\boldsymbol{\alpha}_k, \boldsymbol{S})$, where $\boldsymbol{X}$, $\boldsymbol{Y}$ and $\boldsymbol{S}$ are learnable parameters. The marginalized transition model for the abstract actions is then given by

$$p_{\boldsymbol{\alpha}_k}(l_k|l_{k-1}) = \int p(l_k|l_{k-1}, \boldsymbol{\alpha}_k)p(\boldsymbol{\alpha}_k)d\boldsymbol{\alpha}_k = \mathcal{N}\left(\boldsymbol{X}l_{k-1} + \boldsymbol{Y}\boldsymbol{\mu}_{\boldsymbol{\alpha}_k}, \boldsymbol{S} + \boldsymbol{Y}\boldsymbol{\Sigma}_{\boldsymbol{\alpha}_k}\boldsymbol{Y}^T\right). \tag{7}$$

We can directly use this transition model to obtain the Kalman prediction update which computes the prior belief $p_{\boldsymbol{\alpha}_{1:k}}(l_k|\boldsymbol{\beta}_{1:k-1}) = \mathcal{N}(\boldsymbol{\mu}_{l_k}^-, \boldsymbol{\Sigma}_{l_k}^-)$ from the posterior belief $p_{\boldsymbol{\alpha}_{1:k-1}}(l_{k-1}|\boldsymbol{\beta}_{1:k-1}) = \mathcal{N}(\boldsymbol{\mu}_{l_{k-1}}^+, \boldsymbol{\Sigma}_{l_{k-1}}^+)$ of the previous time window, see Appendix A.

**Observation Update.** Similarly, we will use a linear observation model for the abstract observations $p(\boldsymbol{\beta}_{k,t}|l_k) = \mathcal{N}(\mathbf{H}l_k, \mathrm{diag}(\boldsymbol{\nu}_{k,t}))$ with $\mathbf{H} = [\boldsymbol{I}, \boldsymbol{0}]$. As can be seen from the definition of the observation matrix $\mathbf{H}$, the latent space is also decomposed into its observable and unobservable part, i.e., $l_k = [\boldsymbol{u}_k, \boldsymbol{v}_k]$. In difference to the standard factorized Kalman observation update given in Appendix A, we have to infer with a set of observations $\vec{\beta}_{k,t}$ with $t = 1 \ldots H$ for a single time window $k$. While in principle, the Kalman observation update can be applied incrementally $H$ times to obtain the posterior $p_{\boldsymbol{\alpha}_{1:k}}(l_k|\boldsymbol{\beta}_{1:k}) = \mathcal{N}(\boldsymbol{\mu}_{l_k}^+, \boldsymbol{\Sigma}_{l_k}^+)$, such an update would be very slow and also

cause numerical inaccuracies. Hence, we devise a new permutation invariant version of the update rule that allows parallel processing with set encoders [31]. We found that this update rule is easier to formalize using precision matrices. Hence, we first transform the prior covariance vectors $\boldsymbol{\sigma}_{l_k}^{u-}$, $\boldsymbol{\sigma}_{l_k}^{l-}$ and $\boldsymbol{\sigma}_{l_k}^{s-}$ to its corresponding precision representation $\boldsymbol{\lambda}_{l_k}^{u-}$, $\boldsymbol{\lambda}_{l_k}^{l-}$ and $\boldsymbol{\lambda}_{l_k}^{s-}$ which can be performed using block-wise matrix inversions of $\boldsymbol{\Sigma}_{l_k}^{-}$. Due to the factorization of the covariance matrix, this operation can be performed solely by scalar inversions. As the update equations are rather lengthy, they are given in Appendix A, B. Subsequently, we compute the posterior precision, where only $\boldsymbol{\lambda}_{l_k}^{u}$ is changed by

$$\boldsymbol{\lambda}_{l_k}^{u+} = \boldsymbol{\lambda}_{l_k}^{u-} + \sum_{t=1}^{H} \mathbf{1} \oslash \boldsymbol{\nu}_{k,t} \tag{8}$$

while $\boldsymbol{\lambda}_{l_k}^{l+} = \boldsymbol{\lambda}_{l_k}^{l-}$ and $\boldsymbol{\lambda}_{l_k}^{s+} = \boldsymbol{\lambda}_{l_k}^{s-}$ remain constant. The operator $\oslash$ denotes the element-wise division. From the posterior precision, we can again obtain the posterior covariance vectors $\boldsymbol{\sigma}_{l_k}^{u+}$, $\boldsymbol{\sigma}_{l_k}^{l+}$ and $\boldsymbol{\sigma}_{l_k}^{s+}$ using only scalar inversions, see Appendix A, B. The posterior mean $\boldsymbol{\mu}_{l,k}^{+}$ can now be obtained from the prior mean $\boldsymbol{\mu}_{l,k}^{-}$ as

$$\boldsymbol{\mu}_{l,k}^{+} = \boldsymbol{\mu}_{l,k}^{-} + \left[ \begin{array}{c} \boldsymbol{\sigma}_{l_k}^{u+} \\ \boldsymbol{\sigma}_{l_k}^{s+} \end{array} \right] \odot \left[ \begin{array}{c} \sum_{t=1}^{H} \left( \boldsymbol{\beta}_{k,t} - \boldsymbol{\mu}_{l_k}^{u,-} \right) \oslash \boldsymbol{\nu}_{k,t} \\ \sum_{t=1}^{H} \left( \boldsymbol{\beta}_{k,t} - \boldsymbol{\mu}_{l_k}^{u,-} \right) \oslash \boldsymbol{\nu}_{k,t} \end{array} \right]. \tag{9}$$

Note that for $H = 1$, i.e a single observation, the given equation is equivalent to the RKN updates. Moreover, the given rule constitutes a unification of the batch update rule for Bayesian aggregation [29] and the incremental Kalman update for our factorization of the belief state representation [1] detailed in Appendix A.

### 3.4    A General Definition For an N-level MTS3

An N-level MTS3 can be defined as a family of N-state space models, $\{S_0, S_1, ..., S_{N-1}\}$. Each of the state space model $S_i$ is given by $S_i = (Z_i, A_i, O_i, f_i, h_i, H_i \Delta t, L_i)$, where $Z_i$ is the state space, $A_i$ the action space, and $O_i$ the observation space of the SSM. The parameter $H_i \Delta t$ denotes the discretization time-step and $f_i$ and $h_i$ the dynamics and observation models, respectively. Here, $l_i \in L_i$ is a task descriptor that parametrizes the dynamics model of the SSM and is held constant for a local window of $H_{i+1}$ steps. $l_i$ is a function of the latent state of SSM one level above it, i.e., $S_{i+1}$. The boundary cases can be defined as follows: for $i = 0$, $H_0 = 1$. Similarly, for $i = N - 1$, the latent task descriptor $L_i$ is an empty set. For all $i$, $H_i < H_{i+1}$.

Even though our experiments focus on MTS3 models with 2 hierarchies, extensive experimentation with more hierarchies can be taken as future work.

## 4    MTS3 as a Hierarchical World Model

MTS3 allows for a natural way to build world models that can deal with partial observability, non-stationarity and uncertainty in long-term predictions, properties which are critical for model-based control and planning. Furthermore, introducing several levels of latent variables, each working at a different time scale allows us to learn world models that can make action conditional predictions/"dreams" at multiple time scales and multiple levels of state and action abstractions.

### 4.1    Conditional Multi Time Predictions With World Model

Conditional multi-step ahead predictions involve estimating plausible future states of the world resulting from a sequence of actions. Our principled formalism allows for action-conditional future predictions at multiple levels of temporal abstractions. The prediction update for the sts-SSM makes prior estimates about future latent task variables conditioned on the abstract action representations. Whereas, the task conditional prediction update in the fts-SSM estimates the future prior latent states, conditioned on primitive actions and the inferred latent task priors, which are decoded to reconstruct future observations. For initializing the prior belief $p(\boldsymbol{z}_{k,1})$ for the first time step of the time window $k$, we use the prior belief $p(\boldsymbol{z}_{k-1,H+1})$ of the last time step of the time window $k - 1$.

## 4.2 Optimizing the Predictive Log-Likelihood

The training objective for the MTS3 involves maximizing the posterior predictive log-likelihood which is given below for a single trajectory, i.e.,

$$
\begin{aligned}
L &= \sum_{k=1}^{N} \sum_{t=1}^{H} \log p(\boldsymbol{o}_{k,t+1}|\boldsymbol{\beta}_{1:k-1}, \boldsymbol{\alpha}_{1:k}, \boldsymbol{w}_{k,1:t}, \boldsymbol{a}_{k,1:t}) \\
&= \sum_{k=1}^{N} \sum_{t=1}^{H} \log \iint p(\boldsymbol{o}_{k,t+1}|\boldsymbol{z}_{k,t+1}) p(\boldsymbol{z}_{k,t+1}|\boldsymbol{w}_{k,1:t}, \boldsymbol{a}_{k,1:t}, \boldsymbol{l}_k) p(\boldsymbol{l}_k|\boldsymbol{\beta}_{1:k-1}, \boldsymbol{\alpha}_{1:k}) d\boldsymbol{z}_{k,t+1} d\boldsymbol{l}_k \\
&= \sum_{k=1}^{N} \sum_{t=1}^{H} \log \int p(\boldsymbol{o}_{k,t+1}|\boldsymbol{z}_{k,t+1}) p_{\boldsymbol{l}_k}(\boldsymbol{z}_{k,t+1}|\boldsymbol{w}_{k,1:t}, \boldsymbol{a}_{k,1:t}) d\boldsymbol{z}_{k,t+1}.
\end{aligned}
\tag{10}
$$

The extension to multiple trajectories is straightforward and omitted to keep the notation uncluttered. Here, $\boldsymbol{o}_{k,t+1}$ is the ground truth observations at the time step $t+1$ and time window $k$ which needs to be predicted from all (latent and abstract) observations up to time step $t$. The corresponding latent state prior belief $p_{\boldsymbol{l}_k}(\boldsymbol{z}_{k,t+1}|\boldsymbol{w}_{k,1:t}, \boldsymbol{a}_{k,1:t})$ has a closed form solution as discussed in Section 3.2.

We employ a Gaussian approximation of the posterior predictive log-likelihood of the form $p(\boldsymbol{o}_{k,t+1}|\boldsymbol{\beta}_{1:k-1}, \boldsymbol{\alpha}_{1:k}, \boldsymbol{w}_{k,1:t}, \boldsymbol{a}_{k,1:t}) \approx \mathcal{N}(\boldsymbol{\mu}_{\boldsymbol{o}_{k,t+1}}, \mathrm{diag}(\boldsymbol{\sigma}_{\boldsymbol{o}_{k,t+1}}))$ where we use the mean of the prior belief $\boldsymbol{\mu}_{\boldsymbol{z}_{k,t+1}}^{-}$ to decode the predictive mean, i.e, $\boldsymbol{\mu}_{\boldsymbol{o}_{k,t+1}} = \mathrm{dec}_{\mu}(\boldsymbol{\mu}_{\boldsymbol{z}_{k,t+1}}^{-})$ and the variance estimate of the prior belief to decode the observation variance, i.e., $\boldsymbol{\sigma}_{o_{k,t+1}} = \mathrm{dec}_{\sigma}(\boldsymbol{\Sigma}_{\boldsymbol{z}_{k,t+1}}^{-})$. This approximation can be motivated by a moment-matching perspective and allows for end-to-end optimization of the log-likelihood without using auxiliary objectives such as the ELBO [1].

Gradients are computed using (truncated) backpropagation through time (BPTT) [30] and clipped. We optimize the objective using the Adam [14] stochastic gradient descent optimizer with default parameters. We refer to Appendix A for more details. For training, we also initialize the prior belief $p(\boldsymbol{z}_{k,1})$ with the prior belief $p_{\boldsymbol{l}_{k-1}}(\boldsymbol{z}_{k-1,H+1}|\boldsymbol{w}_{k-1,1:H}, \boldsymbol{a}_{k-1,1:H})$ from of the previous time window $k-1$. However, we cut the gradients for the fast time scale between time windows as this avoids vanishing gradients and we observed a more stable learning behaviour. Yet, the gradients can still flow between time windows for the fts-SSM via the sts-SSM.

## 4.3 Imputation Based Training For Long Term Prediction

Using the given training loss results in models that are good in one-time step prediction but typically perform poorly in long-term predictions as the loss assumes that observations are always available up to time step $t$. To increase the long-term prediction performance, we can treat the long-term prediction problem as a case of the "missing value" problem, where the missing observations are at the future time steps. Thus, to train our model for long-term prediction, we randomly mask a fraction of observations and explicitly task the network to impute the missing observations, resulting in a strong self-supervised learning signal for long-term prediction with varying prediction horizon length. This imputation scheme is applied at both time scales, masking out single-time steps or whole time windows of length H. The imputation mask is also randomly resampled for every mini-batch.

## 5 Related Work

**Multi Time Scale World Models** One of the early works that enabled environment models at different temporal scales to be intermixed, producing temporally abstract world models was proposed by [27]. The work was limited to tabular settings but showed the importance of learning environment dynamics at multiple abstractions. However, there have been limited works that actually solve this problem at scale as discussed in [17]. A probabilistically principled formalism for these has been lacking in literature and this work is an early attempt to address this issue.

**Deep State Space Models.** Deep SSMs combine the benefits of deep neural nets and SSMs by offering tractable probabilistic inference and scalability to high-dimensional and large datasets. [9, 1, 24] use neural network architectures based on exact inference on SSMs and perform state estimation and dynamics prediction tasks. [25] extend these models to modelling non-stationary dynamics.

[16, 13, 10] perform learning and inference on SSMs using variational approximations. However, most of these recurrent state-space models have been evaluated on very short-term prediction tasks in the range of a few milliseconds and model the dynamics at a single time scale.

**Transformers** Recent advancements in Transformers [28, 23, 2], which rely on attention mechanism, have demonstrated superior performance in capturing long-range dependency compared to RNN models in several domains including time series forecasting [33, 18] and learning world models [19]. [33, 18, 20] use transformer architectures based on a direct multistep loss [32] and show promising results for long-term forecasting since they avoid error accumulation from autoregression. On the other hand [19] uses a GPT-like autoregressive version of transformers to learn world models. These deterministic models, however, do not deal with temporal abstractions and uncertainty estimation in a principled manner. Nevertheless, we think Transformers that operate at multiple timescales based on our formalism can be a promising alternative research direction.

# 6 Experiments

In this section, we evaluate our approach to a diverse set of simulated and real-world dynamical systems for long-horizon prediction tasks. Our experiments are designed to answer the following questions. (a) Can MTS3 make accurate long-term deterministic predictions (mean estimates)? (b) Can MTS3 make accurate long-term probabilistic predictions (variance estimates)? (c) How important are the modelling assumptions and training scheme?

## 6.1 Baseline Dynamics Models

While a full description of our baselines can be found in Appendix F, a brief description of them is given here: (a) **RNNs** - We compare our method to two widely used recurrent neural network architectures, LSTMs [11] and GRUs [4]. (b) **RSSMs** - Among several RSSMs from the literature, we chose RKN [1] and HiP-RSSM [25] as these have shown excellent performance for dynamics learning for short-term predictions and rely on exact inference as in our case. (c) **Transformers** - We also compare with two state-of-the-art Transformer [28] variants. The first variant (AR-Transformer) relies on a GPT-like autoregressive prediction [23, 2]. Whereas the second variant (Multi-Transformer) uses direct multi-step loss [32] from recent literature on long horizon time-series forecasting [33, 18, 20]. Here, multistep ahead predictions are performed using a single shot given the action sequences.

## 6.2 Environments and Datasets

We experiment with three broad datasets. While full descriptions of these datasets, dataset creation procedure, and overall statistics are given in Appendix E, a brief description of them is as follows. (a) **D4RL Datasets** - We use a set of 3 different environments/agents from D4RL dataset [6], which includes the HalfCheetah, Medium Maze and Franka Kitchen environment. Each of these was chosen because of their distinct properties like sub-optimal trajectories (HalfCheetah), realistic domains / human demonstrations (Kitchen), multi-task trajectories, non-markovian collection policies (Kitchen and Maze) and availability of long horizon episodes (all three). (b) **Manipulation Datasets** - We use 2 datasets collected from a real excavator arm and a Panda robot. The highly non-linear non-markovian dynamics due to hydraulic actuators in the former and non-stationary dynamics owing to different payloads in the latter make them challenging benchmarks. Furthermore, accurate modelling of the dynamics of these complex systems is important since learning control policies for automation directly on large excavators is economically infeasible and potentially hazardous. (c) **Mobile Robotics Dataset** - We set up a simulated four-wheeled mobile robot traversing a highly uneven terrain of varying steepness generated by a mix of sinusoidal functions. This problem is challenging due to the highly non-linear dynamics involving wheel-terrain interactions and non-stationary dynamics introduced by varying steepness levels. In all datasets, we only use information about agent/object positions and we mask out velocities to create a partially observable setting.

## 6.3 Can MTS3 make accurate long-term deterministic predictions (mean estimates)?

Here we evaluate the quality of the mean estimates for long-term prediction using our approach. The results are reported in terms of "sliding window RMSE" in Figure 3. We see that MTS3 gives consistently good long-term action conditional future predictions on all 6 datasets. Deep Kalman

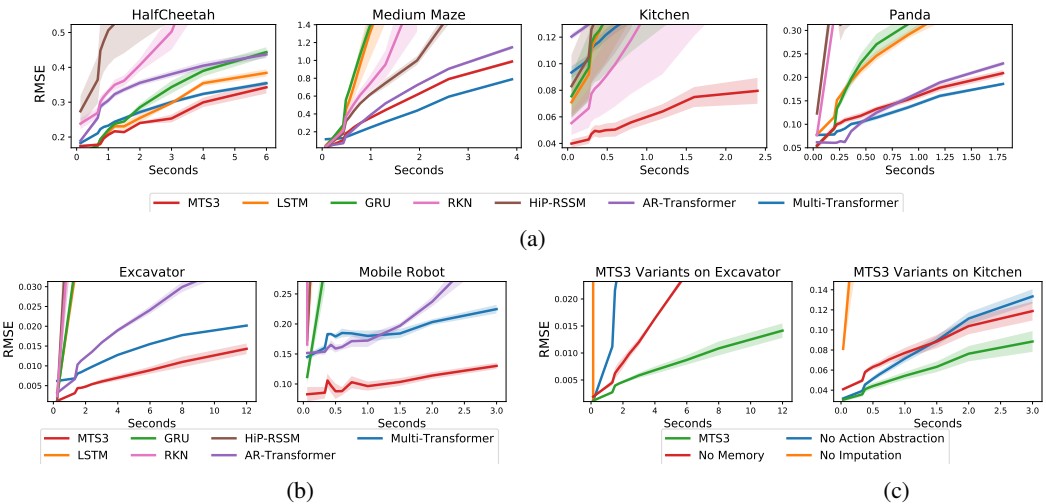

Figure 3: (a) and (b) Comparison with baselines in terms of RMSE for long horizon predictions (in seconds) as discussed in Section 6.3. (c) Ablation with different MTS3 variants is discussed in 6.6.

models [1, 25] which operate on a single time scale fail to give meaningful mean estimates beyond a few milliseconds. Similarly, widely used RNN baselines [11, 4] which form the backbone of several world models [8, 10] give poor action conditional predictions over long horizons. AR-Transformers also fail possibly due to error accumulation caused by the autoregression. However, Multi-Transformers are a strong baseline that outperforms MTS3 in the Medium Maze and Panda dataset by a small margin. However, on more complex tasks like the Kitchen task, which requires modelling multi-object, multi-task interactions [7], MTS3 is the only model that gives meaningful long horizon predictions. A detailed description of the metric "sliding window RMSE" is given in Appendix C. A visualization of the predicted trajectories vs. ground truth is given in Appendix D.

### 6.4 Can MTS3 make accurate long-term probabilistic predictions (variance estimates)?

Next, we examine the question of whether the principled probabilistic inference translates to accurate uncertainty quantification during long-horizon predictions. We trained all the baselines with a negative log-likelihood loss and used the same as a metric to quantify the quality of uncertainty estimates. During the evaluation we rely on a sliding window approach (see Appendix C) and report the results for the last timestep in Table 1. As seen in Table 1, MTS3 gives the most accurate uncertainty estimates in all datasets except Medium Maze, where it is outperformed by Multi-Transformer. Also, notably, AR-Transformers and deep Kalman models fail to learn any meaningful uncertainty representation when it comes to long-term predictions.

| | Prediction Horizon | Algorithm | | | | | | |
|---|---|---|---|---|---|---|---|---|
| | | MTS3 | Multi-Trans | AR-Trans | LSTM | GRU | RKN | HiP-RSSM |
| **Half Cheetah** | 6 s | $-\mathbf{2.80 \pm 0.30}$ | $0.25 \pm 0.05$ | ✗ | $7.34 \pm 0.06$ | $7.49 \pm 0.04$ | ✗ | ✗ |
| **Kitchen** | 2.5 s | $-\mathbf{25.74 \pm 0.12}$ | $-7.3 \pm 0.2$ | ✗ | $32.45 \pm 1.64$ | $32.72 \pm 0.65$ | ✗ | ✗ |
| **Medium Maze** | 4 s | $-0.21 \pm 0.022$ | $-\mathbf{0.88 \pm 0.02}$ | ✗ | $4.03 \pm 0.32$ | $7.76 \pm 0.07$ | ✗ | ✗ |
| **Panda** | 1.8 s | $\mathbf{2.79 \pm 0.32}$ | $3.77 \pm 0.33$ | ✗ | $7.94 \pm 0.39$ | $7.91 \pm 0.23$ | ✗ | ✗ |
| **Hydraulic** | 12 s | $-\mathbf{2.64 \pm 0.12}$ | $-2.46 \pm 0.03$ | ✗ | $7.35 \pm 0.061$ | $7.35 \pm 0.06$ | ✗ | ✗ |
| **Mobile Robot** | 3 s | $-\mathbf{6.47 \pm 0.71}$ | $-5.17 \pm 0.23$ | ✗ | $11.27 \pm 2.3$ | $14.55 \pm 5.6$ | ✗ | ✗ |

Table 1: Comparison in terms of Negative Log Likelihood (NLL) for long horizon predictions (in seconds). Here bold numbers indicate the top methods and ✗ denotes very high/nan values resulting from the highly divergent mean/variance long-term predictions.

### 6.5 How important are the modelling assumptions and training scheme?

Now, we look at three important modelling and training design choices: (i) splitting the latent states to include an unobservable "memory" part using observation model $h^{sts} = h^{fts} = \mathbf{H} = [\mathbf{I}, \mathbf{0}]$ as discussed in Sections 3.3 and 3.2, (ii) action abstractions in Section 3.1.2, (iii) training by imputation.

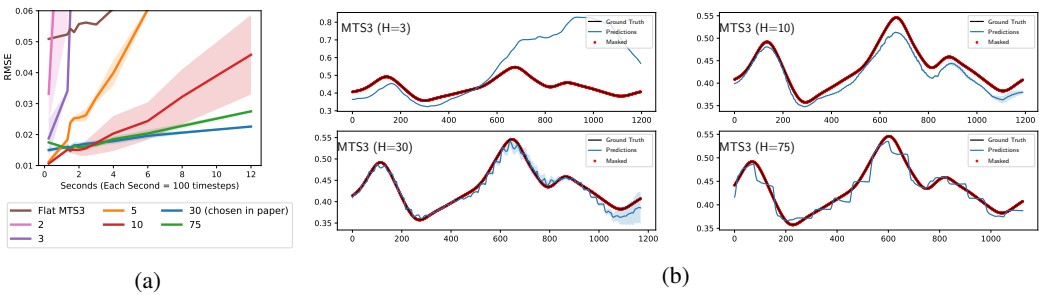

(a)                                                   (b)

Figure 4: Ablation on discretization step $H.\Delta t$ (a) The long-term prediction results in terms of RMSE, with different $H$ values as discussed in Section 6.6 on the hydraulics dataset. (b) The predictions by MTS3 variants with different values of timescale parameter $H.\Delta t$ on a trajectory picked from the hydraulics excavator dataset. The top figures are for $H = 3$ and $H = 10$. Bottom figures are for $H = 30$ and $H = 75$. Note that the results reported in the paper are with $H = 30$.

To analyze the importance of the memory component, we derived and implemented an MTS3 variant with an observation model of $h^{sts} = h^{fts} = \mathbf{I}$ and a pure diagonal matrix representation for the covariance matrices. As seen in Figure 3c, this results in worse long-term predictions, suggesting that splitting the latent states in its observable and unobservable part in MTS3 is critical for learning models of non-markovian dynamical systems. Regarding (ii), we further devised another variant where MTS3 only had access to observations, primitive actions and observation abstractions, but no action abstractions. As seen in our ablation studies, using the action abstraction is crucial for long-horizon predictions.

Our final ablation (iii) shows the importance of an imputation-based training scheme discussed in Section 4.3. As seen in Figure 3c when trained for 1 step ahead predictions without imputation, MTS3 performs significantly worse for long-term prediction suggesting the importance of this training regime.

### 6.6 What is the role of the discretization step $H.\Delta t$?

Finally, we perform ablation for different values of $H.\Delta t$, which controls the time scale of the task dynamics. The results reported are for the hydraulics dataset. The higher the value of H, the slower the timescale of the task dynamics relative to the state dynamics. As seen in Figure 4a, smaller values of $H$ (2,3,5 and 10) give significantly worse performance. Very large values of $H$ (like 75) also result in degradation of performance. To further get an intuitive understanding of the MTS3's behaviour under different timescales, we plot the predictions given by MTS3 for different values of $H$ on a trajectory handpicked from the hydraulics excavator dataset. As seen in Figure 4b, for large values of $H$ like 30 and 75, we notice that the slow-changing task dynamics "reconfigures" the fast dynamics every 30 and 75-step window respectively, by conditioning the lower level dynamics with the newly updated task prior. This effect is noticeable as periodic jumps or discontinuities in the predictions, occurring at 30 and 75-step intervals. Also, for a very large $H$ like 75, the fast time scale ssm has to make many more steps in a longer window resulting in error accumulation and poor predictions.

## 7 Conclusion and Future Work

In this work, we introduce MTS3, a probabilistic formalism for learning the dynamics of complex environments at multiple time scales. By modelling the dynamics of the world at multiple levels of temporal abstraction we capture both the slow-changing long-term trends and fast-changing short-term trends in data, leading to highly accurate predictions spanning several seconds into the future. Our experiments demonstrate that simple linear models with principled modelling assumptions can compete with large transformer model variants that require several times more parameters. Furthermore, our inference scheme also allows for principled uncertainty propagation over long horizons across multiple time scales which capture the stochastic nature of environments. We believe our formalism can benefit multiple future applications including hierarchical planning/control. We discuss the limitations and broader impacts of our work in Appendix G and H.

## 8 Acknowledgement

We thank the anonymous reviewers for the valuable remarks and discussions which greatly improved the quality of this paper. This work was supported by funding from the pilot program Core Informatics of the Helmholtz Association (HGF). The authors acknowledge support by the state of Baden-Württemberg through bwHPC, as well as the HoreKa supercomputer funded by the Ministry of Science, Research and the Arts Baden-Württemberg and by the German Federal Ministry of Education and Research.

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
