# Multi Time Scale World Models

## A    Implementation Details

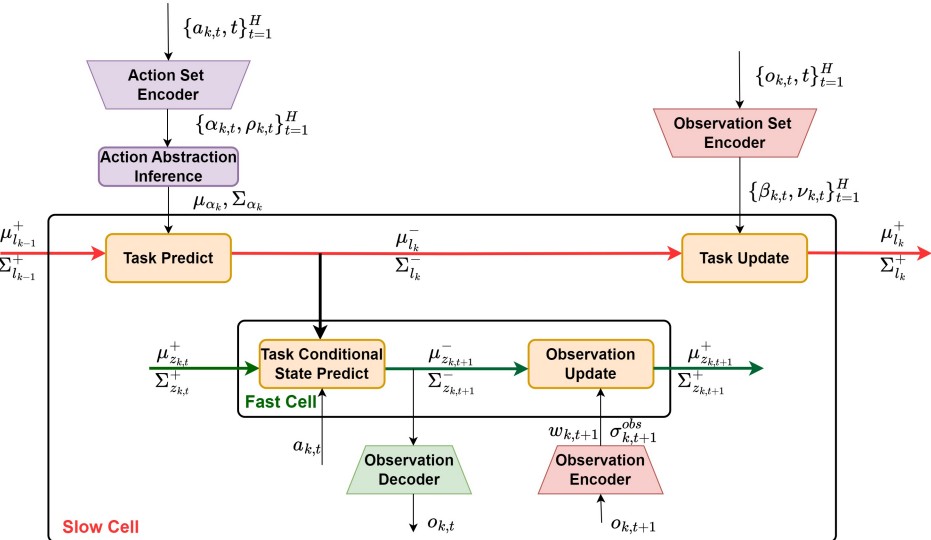

Figure 1: Schematic of a 2-Level MTS3 Architecture. Inference in MTS3 takes place via closed-form equations derived using exact inference, spread across two-time scales. For the fast time scale (fts) SSM, these include the task conditional state predict and observation update stages as discussed in Section 3.2 of the main paper. Whereas, for the slow time scale (sts) SSM, these include the task prediction and task update stages which are described in Section 3.3.

### A.1 Inference In Slow Time Scale SSM

#### A.1.1 Inferring Action Abstraction (sts-SSM)

Given a set of encoded primitive actions and their corresponding variances $\{\boldsymbol{\alpha}_{k,t}, \boldsymbol{\rho}_{k,t}\}_{t=1}^{H}$, using the prior and observation model assumptions in Section 3.1.2 of main paper, we infer the latent abstract action $p(\boldsymbol{\alpha}_k|\boldsymbol{\alpha}_{k,1:H}) = \mathcal{N}(\boldsymbol{\mu}_{\alpha_k}, \boldsymbol{\Sigma}_{\alpha_k}) = \mathcal{N}(\boldsymbol{\mu}_{\alpha_k}, \text{diag}(\sigma_{\alpha_k}))$ as a Bayesian aggregation [10] of these using the following closed-form equations:

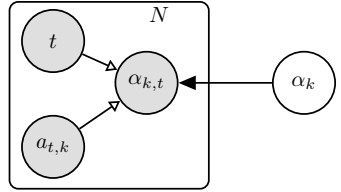

$$\boldsymbol{\sigma}_{\alpha_k} = \left( (\boldsymbol{\sigma}_0)^{\ominus} + \sum_{n=1}^{N} \left( (\boldsymbol{\rho}_{k,t})^{\ominus} \right) \right)^{\ominus},$$

$$\boldsymbol{\mu}_{\alpha_k} = \boldsymbol{\mu}_0 + \boldsymbol{\sigma}_{\alpha_k} \odot \sum_{n=1}^{N} (\boldsymbol{\alpha}_{k,t} - \boldsymbol{\mu}_0) \oslash \boldsymbol{\rho}_{k,t}$$

Figure 2: Generative model for the abstract action $\alpha_k$. The hollow arrows are deterministic transformations leading to implicit distribution $\alpha_{k,t}$ using an action set encoder.

Here, $\ominus$, $\odot$ and $\oslash$ denote element-wise inversion, product, and division, respectively. The update equation is coded as the "abstract action inference" neural network layer as shown in Figure 1.

#### A.1.2 Task Prediction (sts-SSM)

The goal of this step is to update the prior marginal over the latent task variable $\boldsymbol{l}_k$, $p(\boldsymbol{l}_k|\boldsymbol{\beta}_{1:k-1}, \boldsymbol{\alpha}_{1:k})$, given the posterior beliefs from the time window $k-1$ and abstract action $\boldsymbol{\alpha}_k$.

Using the linear dynamics model assumptions from Section 3.3, we can use the following closed-form update equations to compute, $p(\boldsymbol{l}_k|\boldsymbol{\beta}_{1:k-1}, \boldsymbol{\alpha}_{1:k}) = \mathcal{N}(\boldsymbol{\mu}_{l_k}^{-}, \boldsymbol{\Sigma}_{l_k}^{-})$, where

$$\begin{aligned} \boldsymbol{\mu}_{l_k}^{-} &= \mathbf{X}\boldsymbol{\mu}_{l_{k-1}}^{+} + \mathbf{Y}\boldsymbol{\alpha}_k \\ \boldsymbol{\Sigma}_{l_k}^{-} &= \mathbf{X}\boldsymbol{\Sigma}_{l_{k-1}}^{+}\mathbf{X}^T + \mathbf{Y}\boldsymbol{\Sigma}_{\alpha_k}\mathbf{Y}^T + \mathbf{S}. \end{aligned} \tag{1}$$

These closed-form equations are coded as the "task predict" neural net layer as shown in Figure 1.

#### A.1.3 Task Update (sts-SSM)

In this stage, we update the prior over $l_k$ using an abstract observation set $\{\boldsymbol{\beta}_{k,t}\}_{t=1}^{H}$, to obtain the latent task the posterior $\mathcal{N}(\boldsymbol{\mu}_{z_{k,t}}^{+}, \boldsymbol{\Sigma}_{z_{k,t}}^{+}) = \mathcal{N}\left(\begin{bmatrix} \boldsymbol{\mu}_t^{u+} \\ \boldsymbol{\mu}_t^{l+} \end{bmatrix}, \begin{bmatrix} \boldsymbol{\Sigma}_t^{u} & \boldsymbol{\Sigma}_t^{s} \\ \boldsymbol{\Sigma}_t^{s} & \boldsymbol{\Sigma}_t^{l} \end{bmatrix}^{+}\right)$, with $\boldsymbol{\Sigma}_{l_k}^{u} = \text{diag}(\boldsymbol{\sigma}_{l_k}^{u})$, $\boldsymbol{\Sigma}_{l_k}^{l} = \text{diag}(\boldsymbol{\sigma}_{l_k}^{l})$ and $\boldsymbol{\Sigma}_{l_k}^{s} = \text{diag}(\boldsymbol{\sigma}_{l_k}^{s})$.

To do so we first invert the prior covariance matrix $\begin{bmatrix} \boldsymbol{\Sigma}_{l_k}^{u} & \boldsymbol{\Sigma}_{l_k}^{s} \\ \boldsymbol{\Sigma}_{l_k}^{s} & \boldsymbol{\Sigma}_{l_k}^{l} \end{bmatrix}^{+}$ to the precision matrix $\begin{bmatrix} \boldsymbol{\lambda}_{l_k}^{u} & \boldsymbol{\lambda}_{l_k}^{s} \\ \boldsymbol{\lambda}_{l_k}^{s} & \boldsymbol{\lambda}_{l_k}^{l} \end{bmatrix}^{+}$ for permutation invariant parallel processing. The posterior precision is then computed using scalar operations are follows, where only $\boldsymbol{\lambda}_{l_k}^{u}$ is changed by

$$\boldsymbol{\lambda}_{l_k}^{u+} = \boldsymbol{\lambda}_{l_k}^{u-} + \sum_{t=1}^{H} \mathbf{1} \oslash \boldsymbol{\nu}_{k,t} \tag{2}$$

while $\boldsymbol{\lambda}_{l_k}^{l+} = \boldsymbol{\lambda}_{l_k}^{l-}$ and $\boldsymbol{\lambda}_{l_k}^{s+} = \boldsymbol{\lambda}_{l_k}^{s-}$ remain constant. The operator $\oslash$ denotes the element-wise division. The posterior precision is inverted back to the posterior covariance vectors $\boldsymbol{\sigma}_{l_k}^{u+}$, $\boldsymbol{\sigma}_{l_k}^{l+}$ and $\boldsymbol{\sigma}_{l_k}^{s+}$. Now, the posterior mean $\boldsymbol{\mu}_{l,k}^{+}$ can be obtained from the prior mean $\boldsymbol{\mu}_{l,k}^{-}$ as

$$\boldsymbol{\mu}_{l,k}^{+} = \boldsymbol{\mu}_{l,k}^{-} + \begin{bmatrix} \boldsymbol{\sigma}_{l_k}^{u+} \\ \boldsymbol{\sigma}_{l_k}^{s+} \end{bmatrix} \odot \begin{bmatrix} \sum_{t=1}^{H} \left( \boldsymbol{\beta}_{k,t} - \boldsymbol{\mu}_{l_k}^{u,-} \right) \oslash \boldsymbol{\nu}_{k,t} \\ \sum_{t=1}^{H} \left( \boldsymbol{\beta}_{k,t} - \boldsymbol{\mu}_{l_k}^{u,-} \right) \oslash \boldsymbol{\nu}_{k,t} \end{bmatrix}. \tag{3}$$

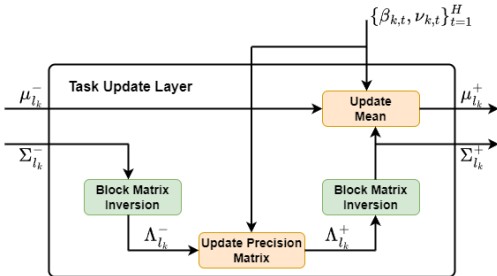

Figure 3: Implementation of task update layer which performs posterior latent task inference in the sts-SSM.

The inversion between the covariance matrix and precision matrix can be done via scalar operations leveraging block diagonal structure as derived in Appendix B. Figure 3 shows the schematic of the task update layer.

## A.2 Inference In Fast Time Scale SSM

The inference in fts-SSM for a time-window $k$ involves two stages as illustrated in Figure **??**, calculating the prior and posterior over the latent state variable $z_t$. To keep the notation uncluttered, we will also omit the time-window index $k$ whenever the context is clear as in section 3.2.

### A.2.1 Task Conditional State Prediction (fts-SSM)

Following the assumptions of a task conditional linear dynamics as in Section 3.2 of the main paper, we obtain the prior marginal for $p(\boldsymbol{z}_{k,t}|\boldsymbol{w}_{1:t-1}^k, \boldsymbol{a}_{1:t-1}^k, \boldsymbol{\beta}_{1:k-1}, \boldsymbol{\alpha}_{1:k-1}) = \mathcal{N}(\boldsymbol{\mu}_{z_{k,t}}^-, \boldsymbol{\Sigma}_{z_{k,t}}^-)$ in closed form, where

$$
\begin{aligned}
\boldsymbol{\mu}_{z_{k,t}}^- &= \mathbf{A}\boldsymbol{\mu}_{z_{k,t-1}}^- + \mathbf{B}\boldsymbol{a}_{k,t-1} + \mathbf{C}\boldsymbol{\mu}_{l_k}^-, \\
\boldsymbol{\Sigma}_{k,t}^- &= \mathbf{A}\boldsymbol{\Sigma}_{k,t-1}^+\mathbf{A}^T + \mathbf{C}\boldsymbol{\Sigma}_{l_k}^-\mathbf{C}^T + \mathbf{Q}.
\end{aligned}
\tag{4}
$$

### A.2.2 Observation Update (fts-SSM)

In this stage, we compute the posterior belief $p(\boldsymbol{z}_{k,t}|\boldsymbol{w}_{1:t}^k, \boldsymbol{a}_{1:t}^k, \boldsymbol{\beta}_{1:k}, \boldsymbol{\alpha}_{1:k-1}) = \mathcal{N}(\boldsymbol{\mu}_{z_{k,t}}^-, \boldsymbol{\Sigma}_{z_{k,t}}^-)$. using the same closed-form update as in [1]. The choice of the special observation model splits the state into two parts, an upper $\boldsymbol{z}_t^{\mathrm{u}}$ and a lower part $\boldsymbol{z}_t^{\mathrm{l}}$, resulting in the posterior belief $\mathcal{N}(\boldsymbol{\mu}_{z_{k,t}}^-, \boldsymbol{\Sigma}_{z_{k,t}}^-) = \mathcal{N}(\begin{bmatrix} \boldsymbol{\mu}_t^{u+} \\ \boldsymbol{\mu}_t^{l+} \end{bmatrix}, \begin{bmatrix} \boldsymbol{\Sigma}_t^u & \boldsymbol{\Sigma}_t^s \\ \boldsymbol{\Sigma}_t^s & \boldsymbol{\Sigma}_t^l \end{bmatrix}^+)$, with $\boldsymbol{\Sigma}_t^u = \mathrm{diag}(\boldsymbol{\sigma}_t^s)$, $\boldsymbol{\Sigma}_t^l = \mathrm{diag}(\boldsymbol{\sigma}_t^l)$ and $\boldsymbol{\Sigma}_t^s = \mathrm{diag}(\boldsymbol{\sigma}_t^s)$. Thus, the factorization allows for only the diagonal and one off-diagonal vector of the covariance to be computed and simplifies the calculation of the mean and posterior to simple scalar operations.

The closed-form equations for the mean can be expressed as the following scalar equations,

$$
\boldsymbol{z}_t^+ = \boldsymbol{z}_t^- + \begin{bmatrix} \boldsymbol{\sigma}_t^{\mathrm{u},-} \\ \boldsymbol{\sigma}_t^{\mathrm{l},-} \end{bmatrix} \odot \begin{bmatrix} \boldsymbol{w}_t - \boldsymbol{z}_t^{\mathrm{u},-} \\ \boldsymbol{w}_t - \boldsymbol{z}_t^{\mathrm{u},-} \end{bmatrix} \oslash \begin{bmatrix} \boldsymbol{\sigma}_t^{\mathrm{u},-} + \boldsymbol{\sigma}_t^{\mathrm{obs}} \\ \boldsymbol{\sigma}_t^{\mathrm{u},-} + \boldsymbol{\sigma}_t^{\mathrm{obs}} \end{bmatrix},
$$

The corresponding equations for the variance update can be expressed as the following scalar operations,

$$
\begin{aligned}
\boldsymbol{\sigma}_t^{\mathrm{u},+} &= \boldsymbol{\sigma}_t^{\mathrm{u},-} \odot \boldsymbol{\sigma}_t^{\mathrm{u},-} \oslash \left( \boldsymbol{\sigma}_t^{\mathrm{u},-} + \boldsymbol{\sigma}_t^{\mathrm{obs}} \right), \\
\boldsymbol{\sigma}_t^{\mathrm{s},+} &= \boldsymbol{\sigma}_t^{\mathrm{u},-} \odot \boldsymbol{\sigma}_t^{\mathrm{s},-} \oslash \left( \boldsymbol{\sigma}_t^{\mathrm{u},-} + \boldsymbol{\sigma}_t^{\mathrm{obs}} \right), \\
\boldsymbol{\sigma}_t^{\mathrm{l},+} &= \boldsymbol{\sigma}_t^{\mathrm{l},-} - \boldsymbol{\sigma}_t^{\mathrm{s},-} \odot \boldsymbol{\sigma}_t^{\mathrm{s},-} \oslash \left( \boldsymbol{\sigma}_t^{\mathrm{u},-} + \boldsymbol{\sigma}_t^{\mathrm{obs}} \right),
\end{aligned}
$$

, where $\odot$ denotes the elementwise vector product and $\oslash$ denotes an elementwise vector division.

### A.3 Modelling Assumptions

### A.3.1 Control Model

To achieve action conditioning within the recurrent cell of fts-SMM, we include a control model $b(a_{k,t})$ in addition to the linear transition model $A_t$. $b(a_{k,t}) = f(a_{k,t})$, where $f(.)$ can be any non-linear function approximator. We use a multi-layer neural network regressor with ReLU activations [8].

However, unlike the fts-SSM where actions are assumed to be known and subjected to no noise, in the sts-SSM, the abstract action is an inferred latent variable with an associated uncertainty estimate. Hence we use a linear control model $Y$, for principled uncertainty propagation.

### A.3.2 Transition Noise

We assume the covariance of the transition noise $Q$ and $S$ in both timescales to be diagonal. The noise is learned and is independent of the latent state.

### A.4 Training

### A.4.1 Training Objective Derivation

We further expand on the training objective in Section 4.2 here. The training objective for the MTS3 involves maximizing the posterior predictive log-likelihood which for a single trajectory, can be derived as,

$$
\begin{aligned}
L &= \sum_{k=1}^{N} \sum_{t=1}^{H} \log p(\boldsymbol{o}_{k,t+1}|\boldsymbol{\beta}_{1:k-1}, \boldsymbol{\alpha}_{1:k-1}, \boldsymbol{w}_{k,1:t}, \boldsymbol{a}_{k,1:t}) \\
&= \sum_{k=1}^{N} \sum_{t=1}^{H} \log \iint p(\boldsymbol{o}_{k,t+1}|\boldsymbol{z}_{k,t+1})p(\boldsymbol{z}_{k,t+1}|\boldsymbol{w}_{k,1:t}, \boldsymbol{a}_{k,1:t}, \boldsymbol{l}_k)p(\boldsymbol{l}_k|\boldsymbol{\beta}_{1:k-1}, \boldsymbol{\alpha}_{1:k-1})d\boldsymbol{z}_{k,t+1}d\boldsymbol{l}_k \\
&= \sum_{k=1}^{N} \sum_{t=1}^{H} \log \int p(\boldsymbol{o}_{k,t+1}|\boldsymbol{z}_{k,t+1})p_{\boldsymbol{l}_k}(\boldsymbol{z}_{k,t+1}|\boldsymbol{w}_{k,1:t}, \boldsymbol{a}_{k,1:t})d\boldsymbol{z}_{k,t+1}.
\end{aligned}
\tag{5}
$$

The extension to multiple trajectories is straightforward. The approximation to the objective is done based on a moment-matching perspective as discussed in Section 4.2 of the main paper.

### A.4.2 Initialization

We initialize the states $\boldsymbol{l}_1$ and $\boldsymbol{z}_{1,1}$ at both timescales for the first-time window $k = 1$ with an all zeros vector and corresponding covariance matrices as $\boldsymbol{\Sigma}_{l_1} = \boldsymbol{\Sigma}_{z_{1,1}} = 10 \cdot \mathbf{I}$. For subsequent windows, the prior belief $p(\boldsymbol{z}_{k,1})$ for the first time step of time window $k$, is initialized using the posterior belief $p_{\boldsymbol{l}_{k-1}}(\boldsymbol{z}_{k-1,H}|\boldsymbol{w}_{k-1,1:H}, \boldsymbol{a}_{k-1,1:H})$ of the last time step of time window $k-1$.

It is also crucial to correctly initialize the transition matrix at both time scales so that the transition does not yield an unstable system. Initially, the transition model should focus on copying the encoder output so that the encoder can learn how to extract good features if observations are available and useful. We initialize the diagonal elements of the transition matrix at both timescales with 1 and the off-diagonal elements with 0.2, while the rest of the elements are set to 0, a choice inspired from [1].

### A.4.3 Learnable Parameters

The learnable parameters in the computation graph are as follows:

**Fast Time Scale SSM:** The linear transition model A, the non-linear control factor b, the linear latent task transformation model C, the transition noise Q, along with the observation encoder and the output decoder.

**Slow Time Scale SSM:** The linear transition model X, the linear control model Y, the transition noise S, along with the observation set encoder and the action set encoder.

# B  Proofs and Derivations

In the following sections vectors are denoted by a lowercase letter in bold, such as "$\mathbf{v}$", while Matrices as an uppercase letter in bold, such as "$\mathbf{M}$". $\mathbf{I}$ denotes identity matrix and $\mathbf{0}$ represents a matrix filled with zeros. For any matrix $\mathbf{M}$, $m$ denotes the corresponding vector of diagonal entries. Also, $\odot$ denotes the elementwise vector product and $\oslash$ denotes an elementwise vector division.

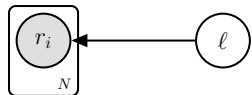

Figure 4:  Graphical Model For Bayesian conditioning with $N$ observations.

## B.1  Bayesian Conditioning As Permutation Invariant Set Operations

**Gaussian Update Rule 1** (Bayesian Conditioning). *Consider the graphical model given in Figure 4, where a set of N conditionally i.i.d observations $\bar{r} = \{r_i\}_{i=1}^{N}$ are generated by a latent variable $l$ and the observation model $p(r_i|l) = \mathcal{N}\left(r_i \mid \mathbf{H}l, diag(\sigma_i^{obs})\right)$. Assuming an observation model $\mathbf{H} = [\mathbf{I}, \mathbf{0}]$, the mean ($\mu$) and precision matrix ($\Lambda$) of the posterior over the latent variable $l$, $p(l|\bar{r}) = \mathcal{N}\left(\mu_l^+, \Sigma_l^+\right) = \mathcal{N}\left(\mu_l^+, (\Lambda_l^+)^{-1}\right)$, given the prior $p_0(l) = \mathcal{N}\left(\mu_l^-, \Sigma_l^-\right) = \mathcal{N}\left(\mu_l^-, (\Lambda_l^-)^{-1}\right)$ have the following permutation invariant closed form updates.*

$$
\begin{aligned}
\Lambda_l^+ &= \Lambda_l^- + \begin{bmatrix} diag(\sum_{i=1}^n \frac{1}{\sigma_i^{obs}}), & \mathbf{0} \\ \mathbf{0}, & \mathbf{0} \end{bmatrix} \\
\mu_l^+ &= \mu_l^- + \begin{bmatrix} \sigma_l^{u+} \\ \sigma_l^{s+} \end{bmatrix} \odot \begin{bmatrix} \sum_{i=1}^{N}\left(r_i - \mu_l^{u,-}\right) \odot \frac{1}{\sigma_i^{obs}} \\ \sum_{i=1}^{N}\left(r_i - \mu_l^{u,-}\right) \odot \frac{1}{\sigma_i^{obs}} \end{bmatrix}
\end{aligned}
\tag{6}
$$

Note that $\Sigma_l$ is the covariance matrix which is the inverse of the precision matrix $\Lambda_l$. Due to the observation model assumption $\mathbf{H} = [\mathbf{I}, \mathbf{0}]$, they take block diagonal form,

$$
\Sigma_l = \begin{bmatrix} \Sigma_l^u & \Sigma_l^s \\ \Sigma_l^s & \Sigma_l^l \end{bmatrix}, \text{ with } \Sigma_u = \text{diag}(\sigma^u), \ \Sigma_l = \text{diag}(\sigma_l^l) \text{ and } \Sigma_s = \text{diag}(\sigma_l^s).
$$

**Proof:**

**Case 1 (Single Observation):**   Before deriving the update rule for $N$ conditionally iid observations, let us start with a simpler case consisting of a single observation $r$. If the marginal Gaussian distribution for the latent variable $l$ takes the form $p(l) = \mathcal{N}\left(l \mid \mu, \Lambda^{-1}\right)$ and the conditional Gaussian distribution for he single observation $r$ given $l$ has the form , $p(\mathbf{r} \mid l) = \mathcal{N}\left(\mathbf{r} \mid \mathbf{H}l + \mathbf{b}, \mathbf{L}^{-1}\right)$. Then the posterior distribution over $l$ can be obtained in closed form as,

$$
p(l \mid \mathbf{r}) = \mathcal{N}\left(l \mid \Sigma\left\{\mathbf{H}^{\mathrm{T}}\mathbf{L}\mathbf{r} + \Lambda\mu\right\}, \Lambda^{-1}\right), \text{ where } \Lambda = \left(\Lambda + \mathbf{H}^{\mathrm{T}}\mathbf{L}\mathbf{H}\right). \tag{7}
$$

We refer to [2] to the proof for this standard result.

**Case 2 (Set Of Observations):**   Now instead of a single observation, we wish to derive a closed form solution for the posterior over latent variable $l \in \mathbb{R}^{2d}$, given a set of N conditionally i.i.d observations $\bar{r} = \{r_i\}_{i=1}^{N}$. Here each element $r_i \in \mathbb{R}^d$ of the set $\bar{r}$ is assumed to to have an observation model $\mathbf{H} = [\mathbf{I}, \mathbf{0}]$. In the derivation, we represent the set of N observations as a random vector

$$
\bar{r} = \begin{bmatrix} r_1 \\ r_2 \\ . \\ . \\ . \\ r_N \end{bmatrix}_{Nd \times 1}.
$$

Since each observation in the set $\bar{r}$ are conditionally independent, we denote the conditional distribution over the context set as $\bar{r} \mid l \sim \mathcal{N}\left(\bar{H}l, \Sigma_r\right)$, where the diagonal covariance matrix has the following form:

$$
\Sigma_r = \begin{bmatrix} \text{diag}(\sigma_{r_1}), & 0, & 0, & .., & 0 \\ 0, & \text{diag}(\sigma_{r_2}), & 0, & .., & 0 \\ ., & ., & ., & .., & . \\ ., & ., & ., & .., & . \\ 0, & 0, & 0, & .., & \text{diag}(\sigma_{r_N}) \end{bmatrix}_{Nd \times Nd}.
$$

The corresponding observation model $\bar{\mathbf{H}}$ is

$$\bar{\mathbf{H}} = \begin{bmatrix} \mathbf{H} \\ \mathbf{H} \\ . \\ . \\ \mathbf{H} \end{bmatrix}_{Nd \times 2d} = \begin{bmatrix} \mathbf{I}, \mathbf{0} \\ \mathbf{I}, \mathbf{0} \\ ., . \\ ., . \\ \mathbf{I}, \mathbf{0} \end{bmatrix}_{Nd \times 2d} .$$

Now given the prior over the latent task variable $l \sim \mathcal{N}\left(\mu_l^-, \Sigma_l^-\right)$, the parameters of the posterior distribution over the task variable, $p(l|\bar{r}) \sim \mathcal{N}\left(\mu_l^+, \Lambda_l^+\right)$, can be obtained in closed-form substituting in Equation (7) as follows.

$$\begin{aligned} \Lambda_l^+ &= (\Sigma_l^+)^{-1} \\ &= \Sigma_l^{-1} + \bar{\mathbf{H}}^T \Sigma_r \bar{\mathbf{H}} \\ &= \Sigma_l^{-1} + \begin{bmatrix} \mathrm{diag}(\boldsymbol{\sigma}_{r_1}), \mathrm{diag}(\boldsymbol{\sigma}_{r_2}), \mathrm{diag}(\boldsymbol{\sigma}_{r_3}), ., ., \mathrm{diag}(\boldsymbol{\sigma}_{r_N}) \\ \mathbf{0}, \quad\quad \mathbf{0}, \quad\quad \mathbf{0}, \quad ., ., \quad \mathbf{0} \end{bmatrix}_{2d \times nd} \bar{\mathbf{H}} \\ &= \boldsymbol{\lambda}_l^- + \begin{bmatrix} \mathrm{diag}(\sum_{i=1}^n \frac{1}{\sigma_{r_i}}), & \mathbf{0} \\ \mathbf{0}, & \mathbf{0} \end{bmatrix}_{2d \times 2d} \end{aligned}$$

$$\begin{aligned} \mu_l^+ &= \mu_l^- + (\Lambda^+)^{-1}\bar{\mathbf{H}}^T \left(\sigma_r^{-2}\mathbf{I}\right)\left(\boldsymbol{y} - \bar{\mathbf{H}}\mu_x\right) & (8) \\ &= \mu_l^- + \Sigma^+ \bar{\mathbf{H}}\left(\sigma_r^{-2}\mathbf{I}\right)\left(\boldsymbol{y} - \bar{\mathbf{H}}\mu_x\right) \\ &= \mu_l^- + \Sigma^+ \begin{bmatrix} \sigma_{r_1}^{-2}\mathbf{I}, \sigma_{r_2}^{-2}\mathbf{I}, \sigma_{r_3}^{-2}\mathbf{I}, ., ., \sigma_{r_n}^{-2}\mathbf{I} \\ \mathbf{0}, \quad \mathbf{0}, \quad \mathbf{0}, \quad ., ., \quad \mathbf{0} \end{bmatrix}\left(\boldsymbol{y} - \bar{\mathbf{H}}\mu_x\right) \\ &= \mu_l^- + \begin{bmatrix} \boldsymbol{\sigma}_l^{u+}, & \boldsymbol{\sigma}_l^{s+} \\ \boldsymbol{\sigma}_l^{s+}, & \boldsymbol{\sigma}_l^{l+} \end{bmatrix}\begin{bmatrix} \sum_{n=1}^N \left(\mathbf{r_n} - \mu_l^{u,-}\right) \odot \frac{1}{\sigma_i} \\ \mathbf{0} \end{bmatrix} \\ &= \mu_l^- + \begin{bmatrix} \boldsymbol{\sigma}_l^{u+} \\ \boldsymbol{\sigma}_l^{s+} \end{bmatrix} \odot \begin{bmatrix} \sum_{i=1}^N \left(\mathbf{r_i} - \mu_l^{u,-}\right) \odot \frac{1}{\sigma_{r_i}} \\ \sum_{i=1}^N \left(\mathbf{r_n} - \mu_l^{u,-}\right) \odot \frac{1}{\sigma_{r_i}} \end{bmatrix} \end{aligned}$$

Here $\mu_l^+$ is the posterior mean and $\boldsymbol{\Lambda}_l^+$ is the posterior precision matrix.

**Corollary 1.** *The closed form updates for the resulting posterior distribution $p(l|\bar{r})$ is permutation invariant with respect to the observation set $\bar{r}$.*

## B.2   Derivation For Matrix Inversions as Scalar Operations

**Inversion Of Block Diagonal Matrix.** *Consider a block matrix of the following form $\boldsymbol{A} = \begin{bmatrix} \mathrm{diag}(\boldsymbol{a}^u) & \mathrm{diag}(\boldsymbol{a}^s) \\ \mathrm{diag}(\boldsymbol{a}^s) & \mathrm{diag}(\boldsymbol{a}^l) \end{bmatrix}$. Then inverse $A^{-1} = \boldsymbol{B}$ can be calculated using scalar operations and is given as, $\boldsymbol{B} = \begin{bmatrix} \mathrm{diag}(\boldsymbol{b}^u) & \mathrm{diag}(\boldsymbol{b}^s) \\ \mathrm{diag}(\boldsymbol{b}^s) & \mathrm{diag}(\boldsymbol{b}^l) \end{bmatrix}$ where,*

$$\begin{aligned} \boldsymbol{b}^u &= \boldsymbol{a}_l \oslash \left(\boldsymbol{a}_u \odot \boldsymbol{a}_l - \boldsymbol{a}_s \odot \boldsymbol{a}_s\right) \\ \boldsymbol{b}^s &= -\boldsymbol{a}_s \oslash \left(\boldsymbol{a}_u \odot \boldsymbol{a}_l - \boldsymbol{a}_s \odot \boldsymbol{a}_s\right) & (9) \\ \boldsymbol{b}^l &= \boldsymbol{a}_u \oslash \left(\boldsymbol{a}_u \odot \boldsymbol{a}_l - \boldsymbol{a}_s \odot \boldsymbol{a}_s\right) \end{aligned}$$

.

**Proof:**   To prove this we will use the following matrix identity of a partitioned matrix from [2], which states

$$\begin{pmatrix} A & B \\ C & D \end{pmatrix}^{-1} = \begin{pmatrix} M & -MBD^{-1} \\ -D^{-1}CM & D^{-1} + D^{-1}CMBD^{-1} \end{pmatrix} \qquad (10)$$

where M is defined as

$$M = \left(A - BD^{-1}C\right)^{-1}.$$

Here M is called the Schur complement of the Matrix on the left side of Equation 10. The algebraic manipulations to arrive at scalar operations in Equation 9 are straightforward.

## C Metrics Used For Measuring Long Horizon Predictions

### C.1 Sliding Window RMSE

The sliding window RMSE (Root Mean Squared Error) metric is computed for a predicted trajectory in comparison to its ground truth. At each time step, the RMSE for each trajectory is determined by taking the root mean square of the differences between the ground truth and predicted values within a sliding window that terminates at the current time step. This sliding window, with a specified size, provides a smoothed localized assessment of prediction accuracy over the entire prediction length. Mathematically, the sliding window RMSE at time step $t$ is given by:

$$\text{RMSE}(t) = \sqrt{\frac{1}{W} \sum_{i=t-W+1}^{t} \left(\text{gt}_i - \text{pred}_i\right)^2}$$

where $t$ is the current time step, $W$ is the window size, and $\text{gt}_i$ and $\text{pred}_i$ are the ground truth and predicted values at time step $i$, respectively. The extension to multiple trajectories is straightforward and omitted to keep the notation uncluttered.

### C.2 Sliding Window NLL

The sliding window NLL (Negative Log-Likelihood) metric is computed for a predicted probability distribution against the true distribution. At each time step, the NLL is determined by summing the negative log-likelihood values within a sliding window that terminates at the current time step. This sliding window, with a specified size, provides a smoothed localized evaluation of prediction accuracy across the entire sequence.

Mathematically, the sliding window NLL at time step $t$ is given by:

$$\text{NLL}(t) = -\frac{1}{W} \sum_{i=t-W+1}^{t} \log \mathcal{N}\left(\text{gt}_i \mid \text{predMean}_i, \text{predVar}_i\right)$$

where $t$ is the current time step, $W$ is the window size. $\text{predMean}_i$, $\text{predVar}_i$, and $\text{gt}_i$ represent the predicted mean, predicted variance, and the ground truth at time step $i$.

## D Additional Experiments and Plots

### D.1 Additional results on ablation with discretization step $H.\Delta t$

In addition to the Hydraulics Dataset discussed in Section 6.4, we report the results of the ablation study with different values of $H.\Delta t$, for the mobile robot dataset. The higher the value of H, the slower the timescale of the task dynamics relative to the state dynamics. As seen in Figure 5, smaller values of H (like 2,3,5 and 10) give significantly worse performance. Very large values of H (like 150) also result in degradation of performance. In the paper, we used a value of H=75.

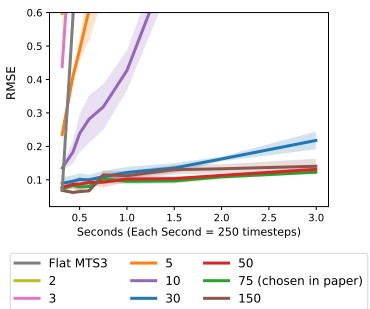

Figure 5: Ablation on discretization step $H.\Delta t$. The long-term prediction results in terms of RMSE, with different $H$ on the mobile dataset.

### D.2 Visualization of predictions given by different models.

In this section, we plot the multistep ahead predictions (mean and variance) by different models on 3 datasets on normalized test trajectories. Not that we omit NaN values in predictions while plotting.

 **D.2.1   Franka Kitchen**

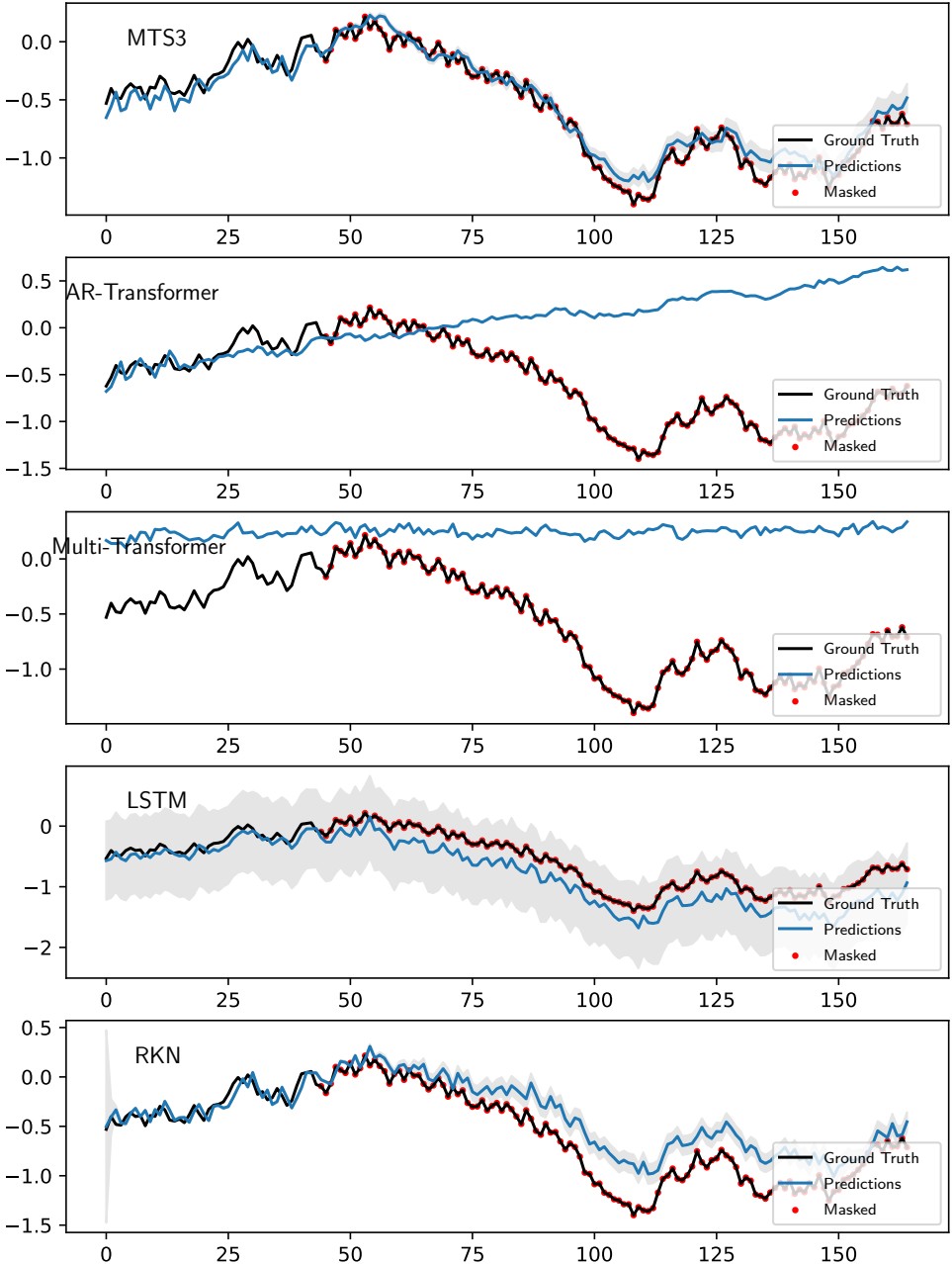

Figure 6: Multi-step ahead mean and variance predictions for a particular joint (joint 1) of Franka Kitchen Environment. The multi-step ahead prediction starts from the first red dot, which indicates masked observations. MTS3 gives the most reliable mean and variance estimates.

 **D.2.2 Hydraulic Excavator**

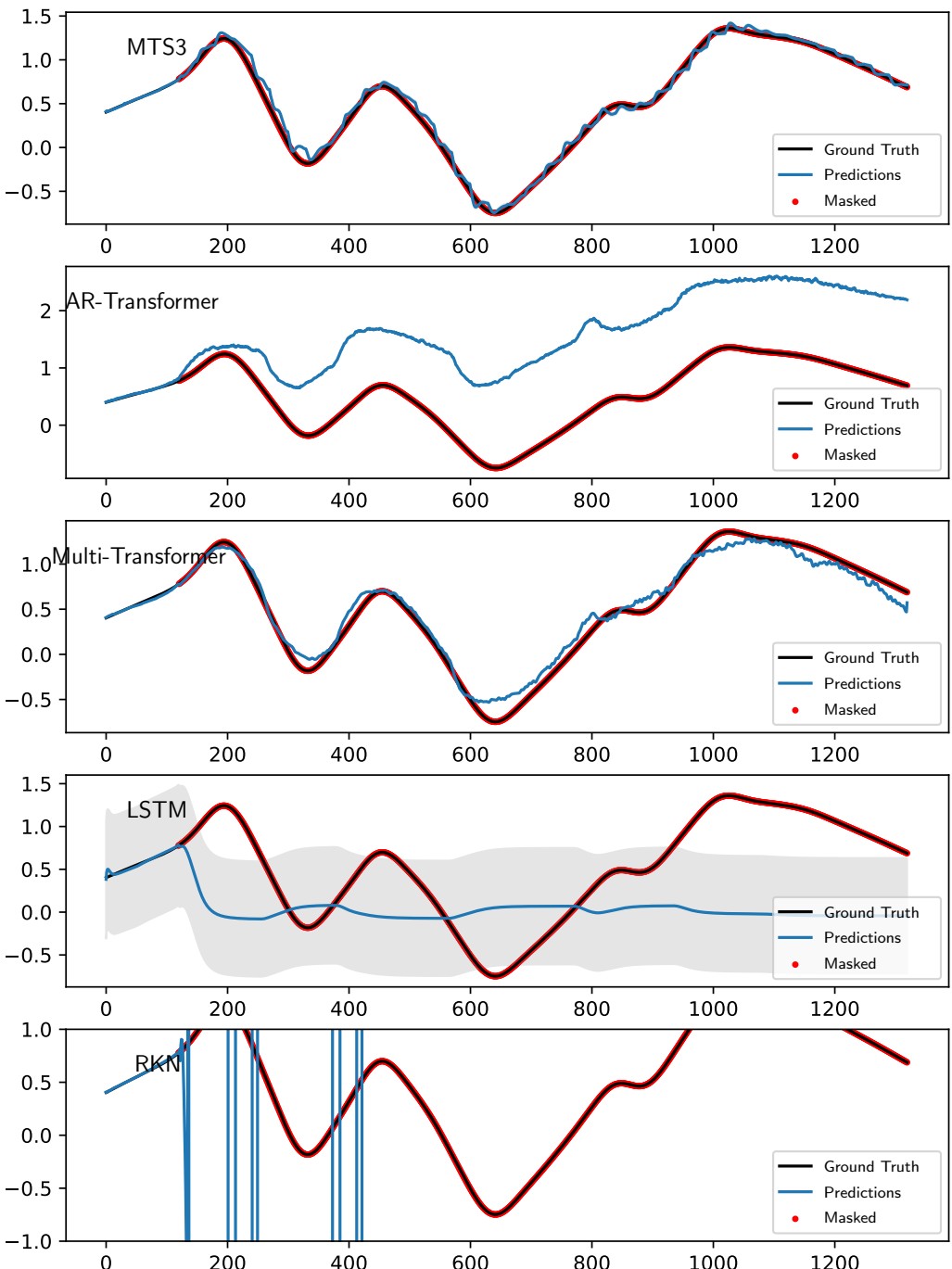

Figure 7: Multi-step ahead mean and variance predictions for a particular joint (joint 1) of Excavator Dataset. The multi-step ahead prediction starts from the first red dot, which indicates masked observations. MTS3 gives the most reliable mean and variance estimates even up to 12 seconds into the future. Another interesting observation can also be seen in the predictions for MTS3, where after every window k of sts-SSM, which is 0.3 seconds (30 timesteps) long, the updation of the higher-level abstractions helps in grounding the lower-level predictions thus helping in the long horizon yet fine-grained predictions.

 **D.2.3 Mobile Robot**

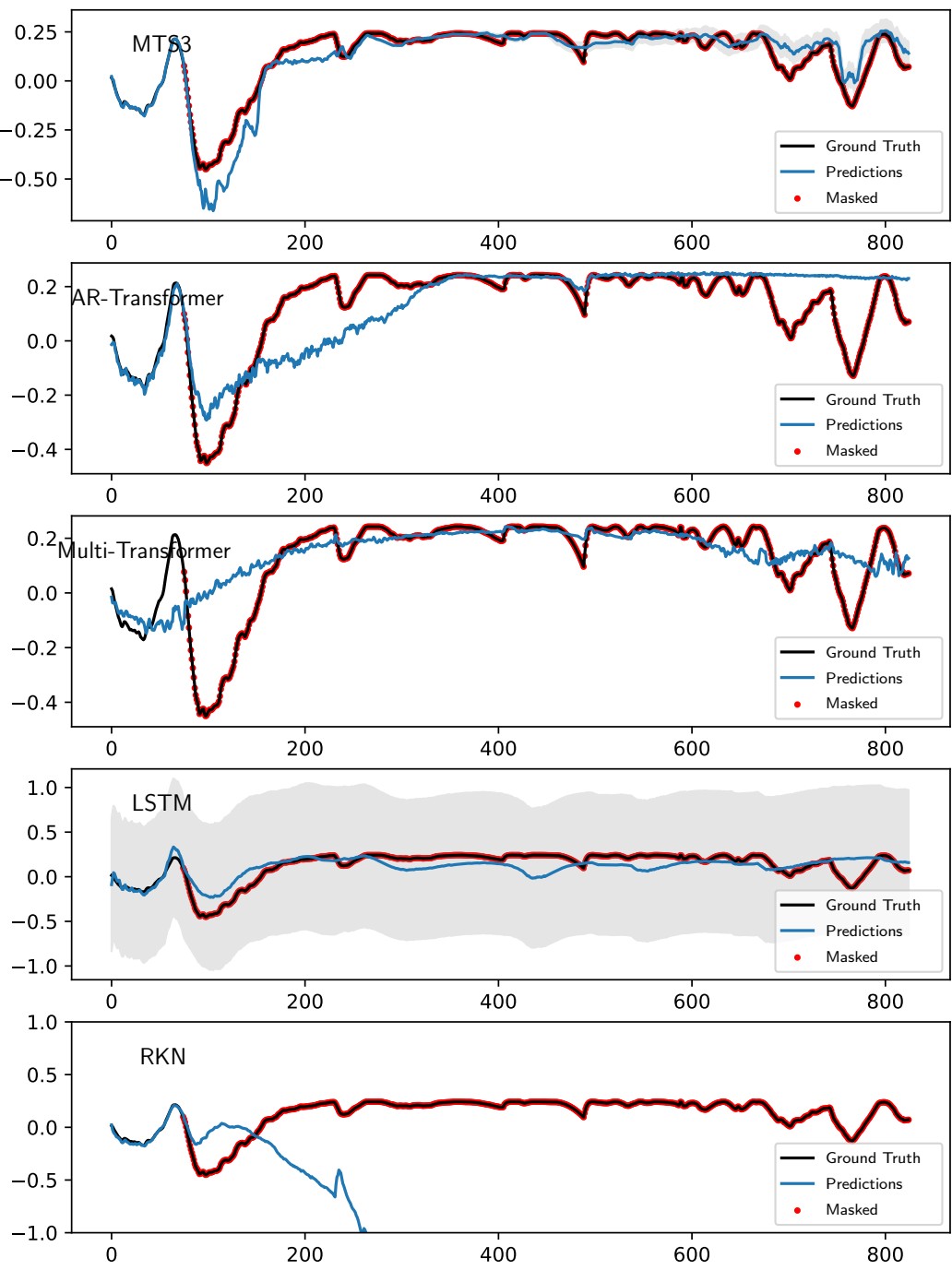

Figure 8: Multi-step ahead mean and variance predictions for a particular joint (joint 7) of Mobile Robot Dataset. The multi-step ahead prediction starts from the first red dot, which indicates masked observations. MTS3 gives the most accurate mean and variance estimates among all algorithms.

## E    Robots and Data

In all datasets, we only use information about agent/object positions and we mask out velocities to create a partially observable setting. All datasets are subjected to a mean zero, unit variance normalization during training. During testing, they are denormalized after predictions. The details of the different datasets used are explained below:

### E.1    D4RL Datasets

**Details:** We use a set of 3 different environments/agents from D4RL dataset [4], which includes the HalfCheetah, Franka Kitchen and Maze2D (medium) environment. **(a) HalfCheetah:** We used 1000 suboptimal trajectories collected from a policy trained to approximately 1/3 the performance of the expert. The observation space consists of 8 joint positions and the action space consists of 6 joint torques collected at 50 Hz frequency. 800 trajectories were used for training and 200 for testing. For the long horizon task, we used 1.2 seconds (60 timesteps) as context and tasked the model to predict 6 seconds (300 timesteps) into the future. **(b) Franka Kitchen:** The goal of the Franka Kitchen environment is to interact with the various objects to reach a desired state configuration. The objects you can interact with include the position of the kettle, flipping the light switch, opening and closing the microwave and cabinet doors, or sliding the other cabinet door. We used the "complete" version of the dataset and collected 1000 trajectories where all four tasks are performed in order. The observation space consists of 30 dimensions (9 joint positions of the robot and 21 object positions). The action space consists of 9 joint velocities clipped between -1 and 1 rad/s. The data was collected at a 50 Hz frequency. 800 trajectories were used for training and 200 for testing. For the long horizon task, we used 0.6 seconds (30 timesteps) as context and tasked the model to predict 2.7 seconds (135 timesteps) into the future. The dataset is complex due to multi-task, multi-object interactions in a single trajectory. **(c) Medium Maze:** We used 20000 trajectories from a 2D Maze environment, where each trajectory consists of a force-actuated ball (along the X and Y axis) moving to a fixed target location. The observation consists of as the (x, y) locations and a 2D action space. The data is collected at 100 Hz frequency. 16000 trajectories were used for training and 4000 for testing. For the long horizon task, we used 0.6 seconds (60 timesteps) as context and tasked the model to predict 3.9 seconds (390 timesteps) into the future. Rendering of the three environments is shown in Figure 9.

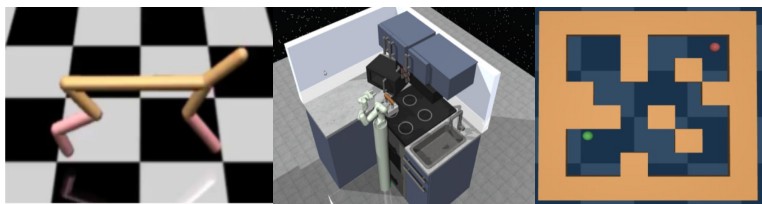

Figure 9: D4RL Environments: (left) HalfCheetah (middle) Franka Kitchen (right) Maze2D-Medium

### E.2    Hydraulic Excavator

**Details:** We collected the data from a wheeled excavator JCB Hydradig 110W show in Figure 10. The data was collected by actuating the boom and arm of the excavator using Multisine and Amplitude-Modulated Pseudo-Random Binary Sequence (APRBS) joystick signals with safety mechanisms in place. A total of 150 mins of data was collected at a frequency of 100 Hz. of which was used as a training dataset and the rest as testing. The observation space consists of the boom and arm positions, while the joystick signals are chosen as actions. For the long horizon task we used 1.5 seconds (150 timesteps) as context and tasked the model to predict 12 seconds (1200 timesteps) into the future.

### E.3    Panda Robot With Varying Payloads

**Details:** We collected the data from a 7 DoF Franka Emika Panda manipulator during free motion and while manipulating loads with weights 0kg (free motion), 0.5 kg, 1 kg, 1.5 kg, 2 kg and 2.5

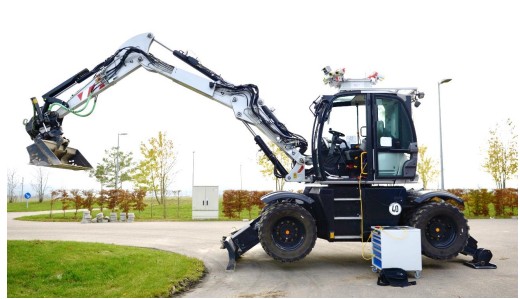
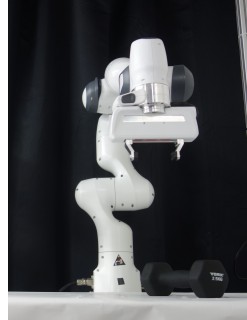

Figure 10: (left) JCB Hydradig 110W Excavator (right) Franka Emika Panda Robot

kg. The robot used is shown in Figure 10. Data is sampled at a frequency of 100 Hz. The training trajectories were motions with loads of 0kg(free motion), 1kg, 1.5kg, and 2.5 kgs, while the testing trajectories contained motions with loads of 0.5 kg and 2 kg. The observations for the forward model consist of the seven joint angles in radians, and the corresponding actions were joint Torques in Nm. For the long horizon task we used 0.6 seconds (60 timesteps) as context and tasked the model to predict 1.8 seconds (180 timesteps) into the future.

## E.4  Wheeled Mobile Robot

**Observation and Data Set:** We collected 50 random trajectories from a Pybullet simulator a wheeled mobile robot traversing terrain with slopes generated by a mix of sine waves as shown in Figure 11. Data is sampled at high frequencies (500Hz). 40 out of the 50 trajectories were used for training and the rest 10 for testing. The observations consist of parameters which completely describe the location and orientation of the robot. The observation of the robot at any time instance $t$ consists of the following features:

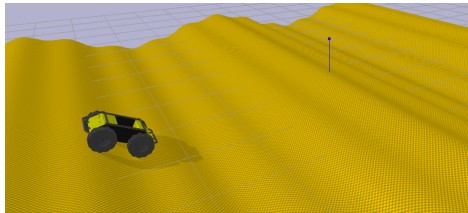

Figure 11: Wheeled Mobile Robot traversing terrain with complex variations in slopes induced by a mix of sine functions.

$$o_t = [x, y, z, \cos(\alpha), \sin(\alpha), \cos(\beta)$$
$$\sin(\beta), \cos(\gamma), \sin(\gamma)]$$

where, $x, y, z$ - denote the global position of the Center of Mass of the robot, $\alpha, \beta, \gamma-$ Roll, pitch and yaw angles of the robot respectively, in the global frame of reference [9]. For the long horizon task we used 0.6 seconds (150 timesteps) as context and tasked the model to predict 3 seconds (750 timesteps) into the future.

## F  Hyperparameters and Compute Resources

**Compute Resources**   For training MTS3, LSTM, GRU and Transformer models we used compute nodes with (i) Nvidia 3090 and (ii) Nvidia 2080 RTX GPUs. For training more computationally expensive locally linear models like RKN, HiP-RSSM we used compute nodes with NVIDIA A100-40 GPUs.

**Hyperparameters**   Hyperparameters were selected via grid search. In general, the performance of MTS3 is not very sensitive to hyperparameters. Among all the baselines, Transformer models were most sensitive to hyperparameters (see Appendix E.5 for details of Transformer architecture).

**Discretization Step:**   For MTS3, the discretization step for the slow time scale SSM as discussed in Section 3.1 for all datasets was fixed as $\mathbf{H} \cdot \Delta t = 0.3$ seconds. In our experiments, we found that discretization values between $0.2 \leq \mathbf{H} \cdot \Delta t \leq 0.5$ seconds give similar performance.

**Rule Of thumb for choosing discretization step in MTS3:**   For any N-level MTS3 as defined in Section 3.4, we recommend searching for discretization factor $H_i$ as a hyperparameter. However, as a general rule of thumb, it can be chosen as $H_i = (\sqrt[N]{T})^i$, where $T$ is the maximum prediction horizon required / episode length. This ensures that very long recurrences are divided between smaller equal-length task-reconfigurable local SSM windows (of length $\sqrt[N]{T}$) spread across several hierarchies.

**Encoder Decoder Architecture:**   For all recurrent models (MTS3, HiP-RSSM, RKN, LSTM and GRU) we use a similar encoder-decoder architecture across datasets. Small variations from these encoder-decoder architecture hyperparameters can still lead to similar prediction performance as reported in the paper.

Observation Set Encoder (MTS3): 1 fully connected + linear output:

- Fully Connected 1: 240, ReLU

Action Set Encoder (MTS3): 1 fully connected + linear output:

- Fully Connected 1: 240, ReLU

Observation Encoder (MTS3, HiP-RSSM, RKN, LSTM, GRU): 1 fully connected + linear output:

- Fully Connected 1: 120, ReLU

Observation Decoder (MTS3, HiP-RSSM, RKN, LSTM, GRU): 1 fully connected + linear output:

- Fully Connected 1: 120, ReLU

Control Model (Primitive Action Encoder) (MTS3, HiP-RSSM, RKN): 1 fully connected + linear output:

- Fully Connected 1: 120, ReLU

The rest of the hyperparameters are described below:

### F.1  D4RL Datasets

#### F.1.1  Half Cheetah

**Recurrent Models**

Transition Model (HiP-RSSM, RKN): number of basis: 32

- $\alpha(\boldsymbol{z}_t)$: No hidden layers - softmax output

| Hyperparameters | MTS3 | HiP-RSSM | RKN | LSTM | GRU |
|---|---|---|---|---|---|
| Learning Rate | 3e-3 | 1e-3 | 1e-3 | 1e-3 | 1e-3 |
| Latent Observation Dimension | 15 | 15 | 15 | 15 | 15 |
| Observation Set Latent Dimension (sts-SSM) | 15 | - | - | - | - |
| Latent State Dimension | 30 | 30 | 30 | 45 | 45 |
| Latent Task Dimension | 30 | 30 | - | - | - |
| Latent Abstract Action Dimension (sts-SSM) | 30 | - | - | - | - |

**Autoregressive Transformer Baseline**
Learning Rate: 1e-5
Optimizer Used: Adam Optimizer
Embedding size: 96
Number of Decoder Layers: 4
Number Of Attention Heads: 4

**Multistep Transformer Baseline**
Learning Rate: 1e-5
Optimizer Used: Adam Optimizer
Embedding size: 128
Number Of Encoder Layers: 2
Number of Decoder Layers: 1
Number Of Attention Heads: 4

### F.1.2 Franka Kitchen

**Recurrent Models**

| Hyperparameters | MTS3 | HiP-RSSM | RKN | LSTM | GRU |
|---|---|---|---|---|---|
| Learning Rate | 3e-3 | 9e-4 | 9e-4 | 1e-3 | 1e-3 |
| Latent Observation Dimension | 30 | 30 | 30 | 30 | 30 |
| Observation Set Latent Dimension (sts-SSM) | 30 | - | - | - | - |
| Latent State Dimension | 60 | 60 | 60 | 90 | 90 |
| Latent Task Dimension | 60 | 60 | - | - | - |
| Latent Abstract Action Dimension (sts-SSM) | 60 | - | - | - | - |

Transition Model (HiP-RSSM, RKN): number of basis: 15

- $\alpha(z_t)$: No hidden layers - softmax output

**Autoregressive Transformer Baseline**
Learning Rate: 5e-5
Optimizer Used: Adam Optimizer
Embedding size: 64
Number of Decoder Layers: 4
Number Of Attention Heads: 4

**Multistep Transformer Baseline**
Learning Rate: 1e-5
Optimizer Used: Adam Optimizer
Embedding size: 64
Number Of Encoder Layers: 2
Number of Decoder Layers: 1
Number Of Attention Heads: 4

### F.1.3 Maze 2D

**Recurrent Models**

Transition Model (HiP-RSSM, RKN): number of basis: 15

| Hyperparameters | MTS3 | HiP-RSSM | RKN | LSTM | GRU |
|---|---|---|---|---|---|
| Learning Rate | 3e-3 | 9e-4 | 9e-4 | 1e-3 | 1e-3 |
| Latent Observation Dimension | 30 | 30 | 30 | 30 | 30 |
| Observation Set Latent Dimension (sts-SSM) | 30 | - | - | - | - |
| Latent State Dimension | 60 | 60 | 60 | 90 | 90 |
| Latent Task Dimension | 60 | 60 | - | - | - |
| Latent Abstract Action Dimension (sts-SSM) | 60 | - | - | - | - |

- $\alpha(z_t)$: No hidden layers - softmax output

**Autoregressive Transformer Baseline**
Learning Rate: 5e-5
Optimizer Used: Adam Optimizer
Embedding size: 96
Number of Decoder Layers: 4
Number Of Attention Heads: 4

**Multistep Transformer Baseline**
Learning Rate: 1e-5
Optimizer Used: Adam Optimizer
Embedding size: 128
Number Of Encoder Layers: 2
Number of Decoder Layers: 1
Number Of Attention Heads: 4

## F.2 Franka Robot Arm With Varying Loads

### Recurrent Models

| Hyperparameters | MTS3 | HiP-RSSM | RKN | LSTM | GRU |
|---|---|---|---|---|---|
| Learning Rate | 3e-3 | 9e-4 | 9e-4 | 3e-3 | 3e-3 |
| Latent Observation Dimension | 15 | 15 | 15 | 15 | 15 |
| Observation Set Latent Dimension (sts-SSM) | 15 | - | - | - | - |
| Latent State Dimension | 30 | 30 | 30 | 45 | 45 |
| Latent Task Dimension | 30 | 30 | - | - | - |
| Latent Abstract Action Dimension (sts-SSM) | 30 | - | - | - | - |

Transition Model (HiP-RSSM,RKN): number of basis: 32

- $\alpha(z_t)$: No hidden layers - softmax output

**Autoregressive Transformer Baseline**
Learning Rate: 5e-5
Optimizer Used: Adam Optimizer
Embedding size: 64
Number of Decoder Layers: 4
Number Of Attention Heads: 4

**Multistep Transformer Baseline**
Learning Rate: 2e-5
Optimizer Used: Adam Optimizer
Embedding size: 64
Number Of Encoder Layers: 2
Number of Decoder Layers: 1
Number Of Attention Heads: 4

## F.3 Hydraulic Excavator

Transition Model (HiP-RSSM,RKN): number of basis: 15

| Hyperparameters | MTS3 | HiP-RSSM | RKN | LSTM | GRU |
|---|---|---|---|---|---|
| Learning Rate | 3e-3 | 8e-4 | 8e-4 | 1e-3 | 1e-3 |
| Latent Observation Dimension | 15 | 15 | 15 | 15 | 15 |
| Observation Set Latent Dimension (sts-SSM) | 15 | - | - | - | - |
| Latent State Dimension | 30 | 30 | 30 | 45 | 45 |
| Latent Task Dimension | 30 | 30 | - | - | - |
| Latent Abstract Action Dimension (sts-SSM) | 30 | - | - | - | - |

317 • coefficient net $\alpha(z_t)$: No hidden layers - softmax output

318 **Autoregressive Transformer Baseline**
319 Learning Rate: 1e-5
320 Optimizer Used: Adam Optimizer
321 Embedding size: 96
322 Number of Decoder Layers: 4
323 Number Of Attention Heads: 4

324 **Multistep Transformer Baseline**
325 Learning Rate: 5e-5
326 Optimizer Used: Adam Optimizer
327 Embedding size: 64
328 Number Of Encoder Layers: 2
329 Number of Decoder Layers: 1
330 Number Of Attention Heads: 4

331 **F.4   Wheeled Robot Traversing Uneven Terrain**

| Hyperparameters | MTS3 | HiP-RSSM | RKN | LSTM | GRU |
|---|---|---|---|---|---|
| Learning Rate | 3e-3 | 8e-4 | 8e-4 | 1e-3 | 1e-3 |
| Latent Observation Dimension | 30 | 30 | 30 | 30 | 30 |
| Observation Set Latent Dimension (sts-SSM) | 30 | - | - | - | - |
| Latent State Dimension | 60 | 60 | 60 | 90 | 90 |
| Latent Task Dimension | 60 | 60 | - | - | - |
| Latent Abstract Action Dimension (sts-SSM) | 60 | - | - | - | - |

332 Transition Model (HiP-RSSM,RKN): number of basis: 15

333 • coefficient net $\alpha(z_t)$: No hidden layers - softmax output

334 **Autoregressive Transformer Baseline**
335 Learning Rate: 5e-5
336 Optimizer Used: Adam Optimizer
337 Embedding size: 128
338 Number of Decoder Layers: 4
339 Number Of Attention Heads: 4

340 **Multistep Transformer Baseline**
341 Learning Rate: 5e-5
342 Optimizer Used: Adam Optimizer
343 Embedding size: 64
344 Number Of Encoder Layers: 4
345 Number of Decoder Layers: 2
346 Number Of Attention Heads: 4

347 **F.5   Transformer Architecture Details**

348 For the AR-Transformer Baseline, we use a GPT-like autoregressive version of transformers except
349 that for the autoregressive input we also concatenate the actions to make action conditional predictions.

For Multi-Transformer we use the same direct multistep prediction and loss as in recent Transformer time-series forecasting literature [12, 6, 7, 11]. A description of the action conditional direct multi-step version of the transformer is given in Algorithm 1.

---

**Algorithm 1:** MultiStep Transformer

---

**Require:** Input past observations $\mathbf{o_{inp}} \in \mathbb{R}^{S \times C}$; Input Past Actions $\mathbf{a_{inp}} \in \mathbb{R}^{S \times A}$; Future
Actions $\mathbf{a_{pred}} \in \mathbb{R}^{O \times A}$; Input Length $S$; Predict length $O$; Observation Dimension $C$; Action
Dimension $A$; Feature dimension $d_k$; Encoder layers number $N$; Decoder layers number $M$.

1: $\mathbf{o_{inp}} \in \mathbb{R}^{S \times C}, \mathbf{a}_{inp} \in \mathbb{R}^{S \times A}, \mathbf{a}_{pred} \in \mathbb{R}^{O \times A}$
2: $\mathbf{X}_{inp} = \text{ConCatFeatureWise}\left(\mathbf{o}_{inp}, \mathbf{a}_{inp}\right)$ $\qquad\qquad\qquad\triangleright\mathbf{X}_{inp} \in \mathbb{R}^{S \times (C+A)}$
3: $\mathbf{X}_{pred} = \text{ConCatFeatureWise}\left(\text{Zeros}(O, C), \mathbf{a}_{pred}\right)$ $\qquad\triangleright\mathbf{X}_{pred} \in \mathbb{R}^{O \times (C+A)}$
4: $\mathbf{X}_{\text{enc}}, \mathbf{X}_{\text{dec}} = \mathbf{X}_{\text{inp}}, \text{ConCat}\left(\mathbf{X}_{inp}, \mathbf{X}_{pred}\right)$ $\quad\triangleright \mathbf{X}_{\text{enc}} \in \mathbb{R}^{S \times (C+A)}, \mathbf{X}_{\text{dec}} \in$
$\mathbb{R}^{(S+O) \times (C+A)}$
5: $\mathbf{X}^0_{\text{enc}} = \text{Embed}\left(\mathbf{X}_{\text{enc}}\right)$ $\qquad\qquad\qquad\qquad\qquad\triangleright\mathbf{X}^0_{\text{enc}} \in \mathbb{R}^{S \times d_k}$
6: **for** $l$ *in* $\{1, \cdots, N\}$ **do**
  7: $\quad\mathbf{X}^{l-1}_{\text{enc}} = \text{LayerNorm}\left(\mathbf{X}^{l-1}_{\text{enc}} + \text{Attn}\left(\mathbf{X}^{l-1}_{\text{enc}}\right)\right)$ $\qquad\triangleright\mathbf{X}^{l-1}_{\text{enc}} \in \mathbb{R}^{S \times d_k}$
  8: $\quad\mathbf{X}^{l}_{\text{enc}} = \text{LayerNorm}\left(\mathbf{X}^{l-1}_{\text{enc}} + \text{FFN}\left(\mathbf{X}^{l-1}_{\text{enc}}\right)\right)$ $\qquad\triangleright\mathbf{X}^{l}_{\text{enc}} \in \mathbb{R}^{S \times d_k}$
**end**
9: $\mathbf{X}^0_{\text{dec}} = \text{Embed}\left(\mathbf{X}_{\text{dec}}\right)$ $\qquad\qquad\triangleright\mathbf{X}^0_{\text{dec}} \in \mathbb{R}^{(S+O) \times d_k}$
10: **for** *for* $l$ *in* $\{1, \cdots, M\}$ **do**
  11: $\quad\mathbf{X}^{l-1}_{\text{dec}} = \text{LayerNorm}\left(\mathbf{X}^{l-1}_{\text{dec}} + \text{Attn}\left(\mathbf{X}^{l-1}_{\text{dec}}\right)\right)$ $\qquad\triangleright$ Decoder
  12: $\quad\mathbf{X}^{l-1}_{\text{dec}} = \text{LayerNorm}\left(\mathbf{X}^{l-1}_{\text{dec}} + \text{Attn}\left(\mathbf{X}^{l-1'}_{\text{dec}}, \mathbf{X}^{N}_{\text{enc}}\right)\right)\triangleright\mathbf{X}^{l-1}_{\text{dec}} \in \mathbb{R}^{(S+O) \times d_k}$
  13: $\quad\mathbf{X}^{l}_{\text{dec}} = \text{LayerNorm}\left(\mathbf{X}^{l-1}_{\text{dec}} + \text{FFN}\left(\mathbf{X}^{l-1}_{\text{dec}}\right)\right)$ $\qquad\triangleright\mathbf{X}^{l}_{\text{dec}} \in \mathbb{R}^{(S+O) \times d_k}$
**end**
14: $\mathbf{y} = \text{MLP}\left(\mathbf{X}^M_{\text{dec}}\right)$ $\qquad\triangleright\mathbf{y} \in \mathbb{R}^{(S+O) \times C}$
15: Return y $\qquad\qquad\qquad\qquad\qquad\qquad\triangleright$ Return the prediction results

---

## G  Limitations

We list some of the limitations of the paper here. (i) We restricted our definition and experiments to MTS3 with two levels of temporal abstractions, which was sufficient in many of our tasks. However, for certain tasks like the Maze2D, we believe more hierarchies can help. As discussed in the main paper the method and inference scheme allows easy addition of more Feudal [3] hierarchies with larger discretization steps ($\mathbf{H} \cdot \Delta t$). (ii) We restrict our application to action conditional long horizon future predictions and do not use the model for (hierarchical) control. A probabilistically principled formalism for hierarchical control as an inference problem, that builds upon MTS3 models is left for future work. (iii) Finally, we restrict our experiments to proprioceptive sensors from the agent and objects. The performance of MTS3 which relies on "reconstruction loss" as the objective is yet to be validated on noisy high dimensional sensor inputs like Images. Image-based experiments and "non-reconstruction" based losses [5] can be taken up as future work.

## H  Broader Impact

While we do not foresee any immediate negative societal impacts of our work, we do believe that machines that can replicate human intelligence at some point should be able to reason at multiple levels of temporal abstractions using internal world models [5]. Having intelligent agents with type 2 reasoning capabilities can have both positive and negative impacts. We believe identifying and mitigating the potentially harmful effects of such autonomous systems is the responsibility of sovereign governments.