# OpenReview forum: "Multi Time Scale World Models"
_NeurIPS.cc/2023/Conference — NeurIPS 2023 spotlight_

### Official Review · Reviewer_hxxZ · 2023-06-21

**Soundness:** 3 good
**Presentation:** 2 fair
**Contribution:** 3 good
**Rating:** 6
**Confidence:** 3

**Summary:**

The paper addresses learning predictive world models that operate at multiple (i.e. 2) time scales. At the slower time scale, the belief over the “task” (i.e. the high-level state) is updated at every H time steps by aggregating the influence of the low-level observations and actions received over that period. The transition dynamics for the fast time scale model resemble those for a standard state space model, with the exception that the they are conditioned upon the high-level state. Throughout, the authors utilise linear transition models in latent space, with Gaussian process and observation noise.

**Strengths:**

* The paper addresses an important problem that is under-explored.
* For the most part the writing is quite clear.
* The experimental results appear strong in comparison to a number of baselines.

**Weaknesses:**

* It is unclear whether the experimental results support the authors’ claim that their model can more accurately capture system dynamics because of the two-time scale approach, or whether their method outperforms the baselines to due to other factors (such as the imputation-based training). An obvious baseline of updating both the low-level and high-level states at the same frequency (i.e. setting H = 1, and therefore removing the multi time scale aspect) is missing. Please see the questions.
* In the preliminaries, the authors imply that they are building upon the locally-linear state space model from [1], that learns linear dynamics that are *conditional upon the state*. However, the authors model the dynamics in latent space to be linear in the latent state, action, and task context (Equation 5), and this linear transition model is *state-independent* (Line 150, i.e. it is the same linear dynamics for all states). It is unclear how the approach can achieve such strong performance with a completely linear model in a fairly low-dimensional latent space. This is further confused by the fact that Section 3.2 is contradicted by Appendix A.3 which instead tells us that the dependence on the actions is in fact non-linear, and learnt by a multi-layer MLP. Please see the questions.
* I think the paper could be stronger if it presented a more general framework (e.g. arbitrary number of hierarchy levels, arbitrary non-linear dynamics (non-Gaussian)), and then presented their 2-timescale, linear Gaussian model as one example of that framework that is computationally simple. Presenting a more general framework would help to set the groundwork for future works that consider the problem of multi time scale world models.

*Minor comments*
- Covariance matrices Q and R do not appear to be defined.
- Line 31: “under non-stationary” words appear to be missing from this sentence

**Questions:**

* How do the results support the case that it is the multi- time scale modelling of your approach that leads to accurate long-term predictions, rather than other factors such as the imputation-based training? From the results, it appears that the imputation-based training scheme (which is a fairly common technique) is crucial for obtaining strong performance. Were the baselines also trained in this manner, or were they only trained to make one-step predictions?
* Can you include an ablation where *H* is set to 1 in your approach (i.e. there is only a single time scale in the model)? This would help to strengthen the case that this work demonstrates that the multiple time scales lead to improved long-term predictions.
* Can the authors explain how they achieve strong results with an entirely linear model as stated in Equation 5? The authors do not appear to use the state-conditional linear model from [1], as they state the linear model is state independent. In principle, we might expect that there exists a high-dimensional latent space where the dynamics are linear, but the authors use fairly low-dimensional latent spaces (e.g. 30 or 60 dimensions). Thus, I find it difficult to understand how the authors approach can outperform methods that allow for non-linear latent dynamics.
* Equation 5 (fully linear) appears to contradict Appendix A.3.1 (non-linear action model). Can you please explain this?
* How should the results in Section 6.4 be interpreted? It is unclear to me how the log-likelihood are supposed to indicate good uncertainty estimates. Are these values the negative log-likelihood of real trajectories evaluated under each model? In which case, wouldn’t the higher likelihood values of MTS3 just indicate that the model is more accurate (not that the uncertainty estimates are better).

I generally like the paper, and I am open to increasing my score if I feel these questions are adequately addressed during the rebuttal period.

**Limitations:**

The limitations section in the appendix is appreciated. A discussion of any limitations associated with the assumption of state-independent linear Gaussian latent dynamics would be useful.

---

> ### Author Rebuttal · Authors · 2023-08-09
>
> We would like to thank the reviewer for posing these critical questions about our model and the suggestions given. We would attempt to answer these with the following paragraphs and global comment (and attached rebuttal pdf document).
>
> **Weakness:** It is unclear whether the experimental results support the authors’ claim that their model can more accurately capture system dynamics because of the two-time scale approach, or other reasons like imputation. **Question:** Can you include an ablation where $H$ is set to 1 in your approach?
>
> We perform ablation for different values of $H$ (everything else remains fixed) in the rebuttal pdf. A value of $H = 1$ resulted in training instabilities and NaN values. The lowest $H$ value that resulted in stable training was 2. As seen in Figure 1 of the rebuttal pdf, smaller $H$ values (relatively fast time scale) for the task levels give significantly worse performance. Furthermore, we devised a "Flat-MTS3" variant with only 1 hierarchy, but all other parameterization and training schemes including imputation kept fixed. This again results in significantly worse performance indicating that it is, in fact, the multi-time scale approach that results in strong empirical results over the baselines.
>
> In the section below we suggest a more systematic way of choosing the $H$ value for an MTS3 with an arbitrary number of hierarchies.
>
> **Weakness** I think the paper could be stronger if it presented a more general framework (e.g. arbitrary number of hierarchy levels, arbitrary...)
>
> We had to limit our definition to 2 timescales because of space constraints. But a definition and details of a generic $N$-level MTS3 is defined in the "Global Comment" section. We also give a general rule of thumb on how the discretization step can be chosen by the rule of thumb (without hyperparameter search) there. We kindly request the reviewer to read the details in the global comments section.
>
> A section for a generic definition of an $N$ Level MTS3 can be included in the final script.
>
> **Question** Were the baselines also trained in this manner or were they only trained to make one-step predictions?
>
> The baselines GRU, LSTM, RKN, HiP-RSSM were in fact trained with imputation schemes as in previous works. Moreover, we also tried a variety of imputation strategies that were not used by the original authors for these models to make them as fair a comparison as possible. However, none of those made much difference for “flat” / single time scale models (deterministic or stochastic). So this also provides evidence that it is, in fact, this multi-time scale formulation that results in a huge improvement in performance.
>
> The ablations with different $H$ in Figure 1 of the rebuttal document and also the prediction plots in Figure 2 further strengthen our claim and we hope this addresses the reviewer's concerns.
>
> **Question** Equation 5 (fully linear) appears to contradict Appendix A.3.1 (non-linear action model). Can you please explain this?
>
> We apologize for causing this confusion and the reviewer is right regarding this. The control model $B$ for the fast ($fts$) SSM is assumed to be non-linear, everything else is fully linear based on insights provided in [7] which studies the effects of action conditioning in great detail. Since the primitive action is known and not uncertain (zero variance), unlike the state transition model $A$ or the task transformation model $C$, we do not need to constrain the model $B$ to be linear as it neither affects the Kalman gain nor the closed-form **covariance** updates in the Kalman predict step.
>
> Also, we think this is a matter of interpretation. The non-linear action model $b$ (mentioned in Appendix), can also be thought of as an encoder which projects the actions to a linear latent space and the matrix $B$ can be interpreted as the identity in this case ($B=I$). Thus, the dynamics still remain fully linear in the latent space.
>
> We will clarify and make a discussion on this in the final version of the paper.
>
> **Weakness:** In the preliminaries, it is unclear ..  strong performance with a completely linear model. **Question:** Can the authors explain how  ... entirely linear model  .. in Equation 5?
>
> We would also like to refer to works [11,12] that marry deep learning and Koopman Theory where finite-dimensional linear embeddings are learned with deep encoders for modeling non-linear dynamics.
>
> However, the real workhorse behind MTS3's surprisingly good results is its capability to model dynamics at multiple temporal abstractions in a top-down fashion with upper levels reconfiguring the lower levels. The multi-time idea has been in discussion in the machine learning community for some time. For eg, the idea of the Hierarchical JEPA architecture proposed (but not implemented) by LeCun in [2] is very similar to what we have in MTS3. We essentially confirmed the hypothesis proposed by LeCun et al. by formalizing and implementing it.
>
> **Question:** How should the results in Section 6.4 be interpreted? ... (not that the uncertainty estimates are better).
>
> We would like to refer to some highly cited literature [8] on how marginal (Negative) Log Likelihood is a combined metric that measures both accuracy and uncertainty estimation of regression tasks. Similar metrics as ours were also used in related literature [9,10] to quantify the uncertainty of probabilistic models. We further plot the uncertainty in predictions (shaded regions) in Appendix C and also the rebuttal pdf (Section 2) to further get an intuitive feel. But we agree that there is still room for improvement in improving the quality of uncertainty estimates and also using better calibration metrics.
>
> We hope the responses here and in the global comment section were able to address most of the questions/concerns raised by the reviewer and we remain hopeful that they would consider raising the score. References are listed in the global comments section.

---

> > ### Comment · Reviewer_hxxZ · 2023-08-10
> >
> > Thanks for the improvements that you have made to the paper during the rebuttal week. My key concerns have been addressed - I think the experiments now more clearly demonstrate the utility of the approach. I will raise my score to a 6.

---

> > > ### Author Response · Authors · 2023-08-13
> > >
> > > Thank you for your reply and for increasing the score!

---

### Official Review · Reviewer_HTHM · 2023-07-02

**Soundness:** 3 good
**Presentation:** 2 fair
**Contribution:** 2 fair
**Rating:** 7
**Confidence:** 3

**Summary:**

The paper proposes a formalized multi-scale world model, which works at two timescales: a fast-timestep module that predicts individual timesteps, and a slower one that is only updated over fixed number of steps. The slower module defines a "task" that controls how the fast module functions, and the slower module reads abstract observations/actions over the time window. The paper derives Bayesian updates and ways to optimize the models. Experiments on three different datasets indicate that the proposed method is significantly better at predicting steps long into the future over baselines, and is also able to better predict the uncertainty.

**Strengths:**

- Formal derivation of the method, including the derivation of the uncertainty bounds.
- Comparison against large number of valid baselines in multiple different settings, and all results indicate the proposed method is better
- Ablation studies on which components of the proposed method are important
- Experiments on both simulators and robots

**Weaknesses:**

- Some weaknesses in the baselines:
	- Only the proposed method had "two-layer" approach timewise, while others methods were "single-layer". While the formal "two-layer" approach was the main selling point of the proposed method, one can also naively create similar effect with RNNs or transformers
		- E.g., see this paper for two-layer RNN in Section 2.1: Jaderberg, Max, Wojciech M. Czarnecki, Iain Dunning, Luke Marris, Guy Lever, Antonio Garcia Castaneda, Charles Beattie et al. "Human-level performance in 3D multiplayer games with population-based reinforcement learning." Science 364, no. 6443 (2019): 859-865.
		- Alternatively, a "single-layer" approach of the proposed method could also be included to better demonstrate the value of "two-layer" approach. However, if two-layer version outperforms single-layer approach, then two-layer results of the other baselines is even more warranted to ensure the benefit is not alone from two-layer approach.
	- Transformers used were rather small (4 layers, ~100 dimension), but many of the previous work has shown that larger transformers perform better (e.g., + 8 layers, 500 dimensions). While larger models are hard to use in robotics (not much compute / tight time constraints for control), testing the scalability of the different methods in terms of parameters to train would provide a better picture of the methods.
		- If larger transformer turns out to be better at world modelling without big impact in inference time, this method still has the benefit of capturing uncertainty, and can potentially be a better fit in planning.
- While paper does mention the potential applications of the world modelling approach, there are no experiments to demonstrate this usefulness. The results seem positive for the proposed method (better at modelling the world), but it is unclear how useful this is down the line. For example, how accurate do you have to be to perform good planning for control?
- No code available. The paper does detail the algorithm and setup used to great detail, but the code might contain details that researchers would need to replicate experiments. Small changes to the underlying libraries or how data is pre-processed may have big effects, or there might be parts in the code that were not reported in the paper. Any code, even if messy, is better than no code.
	- Given the complexity of the algorithm, having at least a pseudocode to refer to is crucial for correct implementation in the future.
	- Example of how code-level implementation details matter: Engstrom, Logan, Andrew Ilyas, Shibani Santurkar, Dimitris Tsipras, Firdaus Janoos, Larry Rudolph, and Aleksander Madry. "Implementation matters in deep rl: A case study on ppo and trpo." In International conference on learning representations. 2019.

**Questions:**

1) Transformer models are known improve in many different tasks as you scale them up in terms of parameters (or: number of layers, dimension size and number of heads per layer). The transformer models used in this paper were rather small (e.g., ~100 dimensionality, few layers). Did you experiment with larger model scales? However, even if transformer models get better at scale, they become slower to infer with, which can be an argument against them while employed in robotics with tight time and compute constraints.
2) How long did it took to train each of the baselines, and how long does it take to run the model (e.g.,, infer steps 5 seconds in to the future)? Knowing the time-to-train and time-to-infer for each method, even if different code was ran on different pieces of hardware. This could strengthen the proposed method's case if it was faster than transformers.

**Limitations:**

Authors acknowledge the limitations of the work, and propose good paths to extend the work. Authors also discuss the broader impact (no immediate societal impact from the work).


## Rebuttal acknowledgement

I have read authors' rebuttal which addressed my concerns, and raised my score from 4 to 7 and confidence from 2 to 3 (before discussion period closed).

---

> ### Author Rebuttal · Authors · 2023-08-09
>
> We thank the reviewer for the comments and suggestions. Here are our replies to some of the weaknesses and questions listed.
>
> **Weakness:** Transformers used were rather small (4 layers, ~100 dimensions), testing the scalability of the different methods in terms of parameters to train. **Questions:** The transformer models used in this paper were rather small (e.g., ~100 dimensionality, few layers). Did you experiment with larger model scales? ... inference time/parameters in robotics with tight time and compute constraints.
>
> Large transformers perform well and generalize when there is no scarcity of data, but it is unfortunately not the case with Robotics. We extensively tested different hyperparameter settings before arriving at this architecture. Larger models resulted in overfitting.
>
> As the reviewer suggested, we are providing an analysis of the number of parameters and inference time for Transformer Vs MTS3 across two datasets in Table 1 in the **rebuttal pdf**. As seen in the table, MTS3 uses a fraction of parameters as transformers.
>
> **Questions:** How long did it take to train each of the baselines, and how long does it take to run the model? ...
>
> We report a comparison for MTS3 vs Transformer baselines for training and inference time in Table 1.  As far as training time is considered, transformers scale better, due to the high parallelization possible at this stage. However, GPT-like Autoregressive transformers used widely in literature take several times more inference time due to autoregression (something not desirable in online robotic deployment). The non-autoregressive Multi-step transformers are faster but have the drawback that it's not flexible as the number of timesteps it can predict ahead is fixed and is hardcoded into the output decoder. Also, note that our code is not optimized as the standard transformer libraries. The performance was evaluated on similar hardware for all models.
>
> **Weakness:** Only the proposed method had a "two-layer" approach timewise, while others methods were "single-layer"... Alternatively, a "single-layer" approach of the proposed method could also be included to better demonstrate the value of the "two-layer" approach.
>
> We include an extensive ablation in the rebuttal pdf, with varying discretization parameters (H) and a "Flat-MTS3" which works at a single time scale as the reviewer suggested. All other aspects of the model were fixed including training regime with imputation. It clearly shows the advantage of learning the dynamics with observation/action abstractions at multiple time scales.
>
> Additionally, we referred to the paper suggested by the reviewer, Jaderberg et al. "Human-level performance in 3D multiplayer games with population-based reinforcement learning." Science 364, no. 6443 (2019): 859-865. Unfortunately, since there is no public codebase available for the paper, we modified our existing LSTM/GRU baselines, with an additional slow-moving LSTM/GRU Cell as shown in Figure 2 of the reference. The paper shows a "bottom to top" architecture, where the low-level states of the fast RNN update the higher-level slow RNN. The hidden states of both RNNs were decoded to make predictions.
>
> Our experiments on both Mobile and Hydraulic robots resulted in worse performance than the single-layer LSTM for this particular architecture. However, we think it would be a fairer comparison only if we have access to their actual codebase or more architectural details.
>
> Also, the 2-layer bottom-up RNN in the said reference was designed to capture the non-markovian state representation in the context of model-free RL policies, and we do not think this was meant for multi-step ahead predictions. We think that a Top to Bottom hierarchy for the latent variables as in our case is essential for making long-term predictions. Ideally, the model should first make the easier higher-level task predictions (more abstract information without worrying about too many details as in lower levels), independent of the lower-level tasks and use the higher-level predictions to condition predictions on lower-level windows. MTS3 maintains this top-to-bottom causal relationship between latent variables at multiple temporal abstractions in a principled way.
>
> We would also request the reviewer to take a look at the global comment section, where we place our work in a different light. We think principled formalisms (e.g., SSMs, MDPs, Semi-MDPs in hierarchical RL context) are important for algorithmic advancements, and we lay a foundation for such a formalism for learning models of the world at multiple temporal abstractions.
>
> **Weakness:** No Code Available
>
> We submit a link to an unofficial version of our codebase [here](https://drive.google.com/file/d/103fbLz1ahrfbSCGN0Ko58A7v76EVLPlu/).
>
> We hope we were able to provide sufficient answers to the questions posted by the reviewer and hope that the reviewer would look at the work in a positive light. Please let us know if you need any further clarifications.

---

> > ### Comment · Reviewer_HTHM · 2023-08-14
> >
> > Thank you for you answers! I especially enjoyed the explanation and clarification of top-to-bottom vs. bottom-to-top approach and differences between the paper I linked, which I do see as not being compatible here. I also agree with the view that this provides more theoretical fundamental ground for world-models, and the exact implementation with RNNs/Transformers/whatnots is future work that build on this.
> >
> > I have one additional question: how was the evaluation done, exactly? This is how I understand it:
> >
> > 1) Take a real trajectory from the dataset of length N + M
> > 2) Take the first N timesteps, and provide these as ground-truth context to the model
> > 3) Model predicts the remaining timesteps M
> > 4) Compute RMSE between true timesteps in the trajectory vs. predicted steps
> >
> > Is this correct? How were the robot actions handled? Were they predicted as part of the model, or do you provide the real actions from the trajectory? If I understood the description right, only states are predicted, but I am asking just to be sure.

---

> > > ### Author Response · Authors · 2023-08-14
> > >
> > > Thank you for the reply to the rebuttal and your understanding.
> > >
> > > The reviewer is right regarding the evaluation setup. We collect a ground truth trajectory of observations and actions of length (N + M), from an agent. During inference with MTS3, the first N observations are given as input context (to observation/task encoders), while the rest of the M observations are masked. The model is now tasked to decode N + M observations, conditioned on N + M **known** action/control inputs.
> > >
> > > We calculate the RMSE between the predicted/decoded M observations and ground truth observations.
> > >
> > > Only observations are predicted/decoded and not actions as they are known (we make action-conditional future observation predictions).
> > >
> > > We hope this answers your question! Please let us know if you need any more clarifications.

---

> > > > ### Comment · Reviewer_HTHM · 2023-08-15
> > > >
> > > > Thank you for the clarification! This and other replies address all my concerns I had, and some additional ones in other comments. I have increased my score from 4 to 7 to clearly signal my vote to have this work accepted (and confidence from 2 to 3). The work does lay a more foundational framework to do world modelling, and then shows strong results across different environments against multiple baselines. I believe this work is of interest to researchers working on world modelling, and as such should be accepted.
> > > >
> > > > I just have one request to authors: open source your experiment code (and data and models) as extensively as you can. I realize this is your intention as is, but I want to highlight how important this is. Without code, others can not realiably replicate the results and the work may get easily forgotten. Even better, if you provide the trained models, others can better study how they behave, where they succeed and where they fail.

---

> > > > > ### Author Response · Authors · 2023-08-19
> > > > >
> > > > > Many thanks to the reviewer for the positive comments on the paper and the increased score... Thank you for the further suggestions... We will be open-sourcing an official user-friendly version of the code and data with proper documentation so that other interested researchers can study the model and build upon the idea/formalism...

---

### Official Review · Reviewer_BLVQ · 2023-07-06

**Soundness:** 3 good
**Presentation:** 3 good
**Contribution:** 3 good
**Rating:** 6
**Confidence:** 4

**Summary:**

Looking to tackle the lack of temporal granularity in existing world models, the paper proposes a multi-time scale linear Gaussian state space model (MTS3). The model uses an efficient closed-form inference scheme on multiple time scales for highly accurate long-horizon predictions and uncertainty estimates over longer horizons. The experiments focus on action conditional long horizon future predictions, showing that MTS3 outperforms recent methods on several system identification benchmarks, including the D4RL dataset, a simulated mobile robot, and real manipulators including data from heavy machinery excavators.

**Strengths:**

-	Multi-Time Scale Predictions: Training scalable hierarchical world models that operate at multiple timescales is an open challenge in multiple fields including compressed video prediction, reinforcement learning, control theory, etc. The paper presents one way to interleave between different levels conditioned on a task descriptor. The low-level learns the dynamics conditioned on a particular task and the higher-level is trained to predict the next task.
-	Efficient Inference Scheme via factorised formulation: Following Becker, et al, 2019, and Volpp, et al 2020, the paper proposes multi-scale inference via closed-form solutions using simplifying locally linear assumptions.
-	I like the way the two levels are connected: through the specification of the prior belief $p(l_k| B_{1:k-1}, \alpha_{1:k-1})$ that defines the $p(l_k)$ in the fast-time scale.
-	The use of a probabilistic formalism allows the model to handle uncertainty in predictions, particularly in prediction of changes to change, which is a common challenge across continual learning task settings.
-	Robust Performance: The model has been shown to outperform recent methods on several system benchmarks for long horizon predictions as noted in the figure 3.
-	Captures Complex Dynamics: The model can better capture the complex, non-linear dynamics of a system in a more efficient and robust way than models that learn on a single time scale (Fig. 5)


**Weaknesses:**

-	In the formulation, the slow time scale SSM is only updated every step, i.e., the slow time $H$ scale time step is given by $H \triangle t$. Therefore, the results are contingent on a design choice. I am cognizant that the this was evaluated for the range $0.2 – 0.5 $. However, how dramatically could results / predictions change if the wrong discretization step was chosen. Is there a way to systematically infer this through the system?
-	The results have presented do not show how well the prediction settings vary across different tasks.
-	The results haven’t shown how well large deviances in the dynamics would be encoded? Example some large spike.
-	Higher level encoded space is based on the observation space. Therefore, the higher-level is inherently conditioned on the first level. So, how would the scaling up work for multiple levels? Would $o \to B$ change to $o \to z \to B$?


**Questions:**

Conceptual:

-	Both abstractions interrelate with each other in the sense that the higher-level predictions can be turned into low-level moment-by-moment predictions? How closely aligned where the encoded spaces across the levels for k-1, k when considering transition from t at k-1 to the next task?
-	It is stated that it is trivial to extend to arbitrary number of temporal abstractions – would it be possible how this would work when there aren’t explicit factors like task that can determine the timestep H?
-	How similar do the tasks, $l$ have to be for the results to hold? Change in environment dynamics? It was not clear to me how well the dynamics are encoded as the model transitions predictions from l to l+n, etc.
-	Is there a reason why the observation abstraction was chosen over the state space?

Formulation, results and presentation:

-	How is the second half of the latent state $d_t$ defined as the derivative that the model uses to the estimate the change of the observable part? Would this work for image-based observations formulations of the same model.
-	The posterior calculation $P(z_t|o,a)$ involves a factorised over the covariance matrix. Where are $s, l$ defined?
-	In eq1. How are Q and R parameterised; are these the $diag \sigma_{z;o}?
-	What happens when H is manipulated?
-	Can tasks from different environments be learnt? Or would this be non-trivial?
-	What does the temporal encoding for both encoding of abstracted observation and action look like? Is this the number of steps taken or latent encoding of it?
-	Would it be possible to clarify what $m_t$ action entails; is this the derivative or something else?
-	The prior belief for the first time step of time window is initialised using the posterior belief of the last time step of time window; does this work for instances there is a massive shift in dynamics?

Minor:

-	Is there a reason why some equations are numbered and others not?


**Limitations:**

The authors have appropriately addressed the limitations of MTS3, specifically regarding the consideration of only two timescales and the model's exclusive evaluation for predictions. Typically, it is customary to validate methods in a controlled setting when assessing the performance of the world model. However, I appreciate the learning the policy from such formulations is tricky and requires further consideration.

---

> ### Author Rebuttal · Authors · 2023-08-09
>
> We thank the reviewer for the positive feedback and valuable suggestions. We write our replies to the weaknesses/questions listed by the reviewer below.
>
> **(Weakness) "However, how dramatically could results ... wrong discretization step chosen. Is there a way to systematically infer this through the system?"**
>
> Thank you for your insightful comment. The discretization step is an important hyperparameter. We chose $0.2s \leq H \Delta t \leq 0.5s$, based on related works in meta-learning[3], HiP-MDP / HiP-SSM [4], where for most physical systems like robots, dynamics can be assumed to be fixed for such a short duration.
>
> However, we further performed an ablation and some intuitive prediction plots for different $H$ on the mobile robot and hydraulic excavator dataset. The results of this can be found in **the rebuttal pdf**. We mention a rule of thumb on how to choose $H$ in different levels in the next section.
>
> **(Question) It is stated that it is trivial to extend to an arbitrary number of temporal abstractions – would it be possible how this would work when there aren’t explicit factors like tasks that can determine the timestep $H$?**
>
> We had to limit our definition to 2 timescales because of space constraints. But a definition and details of a generic N-level MTS3 is defined in the "Global Comment" section. We also give a general rule of thumb on how the discretization step can be chosen (without hyperparameter search) there. We kindly request the reviewer to read the details in the global comments section.
>
> **(Weakness) Higher level encoded space is based on the observation space...  how would the scaling up work for multiple levels? Would it be o -> B or o -> z -> B ?**
> **(Question) Is there a reason why the observation abstraction was chosen over the state space?**
>
> We hope the general definition of MTS3 and the computational/implementation details answer the question on how scaling up would work. We chose this approach to maintain a Feudal [5] (Top to Bottom) hierarchy for the latent variables. Ideally, the model should first make the easier higher-level task predictions (more abstract information without worrying about too many details as in lower levels), independent of the lower-level tasks and use the higher-level predictions to condition predictions on lower-level windows. Using observation abstractions allows for this feudal causal relationship.
>
> **(Question) How is the second half of the latent state $d_t$... Would this work for image-based observations .. ?**
>
> Choosing the observation model as $H=[I,0]$ allows for this division of the latent state vector into two distinct components. The upper part utilizes the identity matrix $I$ in $H$ to directly extract information from the observations. Meanwhile, the second lower part remains unobservable and is meant to hold information inferred over time, such as velocities in ordinary dynamical systems or images. The key aspect that contributes to the effectiveness of this choice is the selection of the covariance matrix structure (as mentioned in line 65 of the paper). The covariance matrix is designed to incorporate both diagonal and off-diagonal elements, ensuring that the correlation between the memory and the observation parts is effectively learned in the off-diagonal part. On the contrary, if we were to utilize a pure diagonal covariance structure, it would not update the memory units (the later half) or their variance adequately during the Observation/Task/Kalman Update step.
>
> Yes, this assumption can effectively deal with image-based observations too as demonstrated by [6].
>
> **(Question) How are Q and R parameterized?**
> We apologize for not making this clear in the paper. The transition noise $Q$ is assumed to be diagonal. It is learned and is independent of the state following [6]. The observation noise $R$ is again diagonal and is the output of the observation encoder, which is used in the Kalman Update / Bayesian Conditioning stage. This would be clarified and made more evident in the final version.
>
> **(Question) What happens when $H$ is manipulated?**
> Answered above.
>
> **(Question) Can tasks from different environments ... be non-trivial?**
> The challenge here would be dealing with varying dimensional observation and action space as the agent/environment changes. If this challenge can be addressed, MTS3 should scale to different environments in our opinion by learning environment-specific temporal abstractions for adaptation.
>
> **(Question) What does the temporal encoding ...?**
> It is the normalized value of the absolute number of steps taken in a particular window (the absolute number of steps in a window is always $1 \leq t \leq H$). We didn't use any encoding.
>
> **(Question) Would it be possible to clarify what $m_t$ action entails; is this the derivative or something else?**
> Could the reviewer clarify which notation in the manuscript they are referring to? We would be happy to clarify this.
>
> **(Weakness / Question): How well large/massive shift in dynamics would be encoded? For example some large spike.**
> We understand that since we have Gaussian assumptions, it's natural to pose this question about discrete changes. We can't learn/predict changes in dynamics caused by external rare events that are not predictable, like a spike (up and down suddenly) due to a human pushing a robot for a brief duration unexpectedly. However, we do think it can handle dynamics changes in the form of step functions (e.g., the load of the robot changes at set durations etc) as long as these changes have a pattern. The Franka Kitchen environment and the Panda real robot encounter such scenarios in our experiments.
>
> We hope to have addressed most of the questions that the reviewer pointed out here and in global comments. We thank the reviewer again for their insights and remain hopeful that they would consider increasing the score.
>
> Please refer to the global comment section for references.

---

> > ### Comment · Reviewer_BLVQ · 2023-08-21
> >
> > Thank you for your response, I really appreciate it. I will maintain my positive opinion of the paper.

---

> > > ### Author Response · Authors · 2023-08-22
> > >
> > > Many thanks for your reply and positive feedback!

---

> ### Comment · Area_Chair_aByi · 2023-08-17
>
> Dear Reviewer,
>
> The author has posted their rebuttal, but you have not yet posted your response. Please post your thoughts after reading the rebuttal and other reviews as soon as possible. All reviewers are requested to post this after-rebuttal-response.

---

### Official Review · Reviewer_tb9F · 2023-07-06

**Soundness:** 4 excellent
**Presentation:** 3 good
**Contribution:** 3 good
**Rating:** 7
**Confidence:** 3

**Summary:**

The authors introduce the Multi Time Scale State Space (MTS3) model in this work. The model uses closed-form equations derived using exact inference, spread across two time-scales, to produce long-horizon predictions and uncertainty estimates. They demonstrate the superiority/competitiveness of their inference approach across a number of offline datasets, both in terms of long-term deterministic predictions, and long-term uncertainty quantification.

**Strengths:**

- To the best of my knowledge, this closed-form multi time-scale inference approach using SSMs is novel, and has not be explored in previous works. The produced inference model is a principled (but non-trivial) integration of pre-existing components, and the results demonstrate the promise of such an approach
- Generally, the work is of high quality. The writing is clear, and the paper is well-organized. The related work section is brief, but appears to adequately address prior work relevant to the aims of this paper.

**Weaknesses:**

- In the Figure 3 ablation plots, MTS3 should remain as Red in (c) to improve clarity. It would also be much clearer if the ablations were displayed in their own figure. It also does not make much sense, beyond organization of the plots onto separate lines, why Figure 3a and 3b are separated, as they are displaying the same findings across different environments. The a/b grouping contains no semantic difference. I understand the need for conserving space, but the way the figures are grouped and color-coded, it is not clear at a first glance what the relationship between a, b, and c are.
- Spacing between Table 1 and the caption is too tight.
- Figure 1 should be raised so that the caption does not overlap with 3.1 (try to align with line 79 paragraph).

**Questions:**

- How were the hyperparameters for MTS3 tuned? How were the baselines tuned?
- Transformers are achieving impressive results in control as of late. Where might they be integrated into this work in the future, to improve scalability, generalization, etc.?

**Limitations:**

- The modeling assumptions constitute most of the limitations, but these are also what enable the convenient closed-form updates.
- The authors note that the their model is limited to two levels of temporal abstraction, and that for certain tasks (e.g. Maze2D) more hierarchies may help. The method allows for addition of more complex abstractions such as Feudal hierarchies.
- The authors are restricting their application to action conditional long-horizon future predictions. Although they note that future work can use these predictions for hierarchical control, they leave this for future work—it would have been nice to demonstrate this possibility with real control results.
- Their method relies on reconstruction loss, which may have limited direct application to image-based domains. However, as the authors note, non-reconstruction based losses can be integrated. Again, it would have been nice to see this integration of different losses directly.
- Overall, while I believe the authors have extensively noted the major limitations, especially for listed limitations (ii) and (iii), it would have been nice to see results in this paper to demonstrate that these limitations can indeed be relieved with simple substitution/introduction of new components.

---

> ### Author Rebuttal · Authors · 2023-08-09
>
> We thank the reviewer for looking at our submission in a positive light. We would like to address a few questions raised by the reviewer below:
>
> **Question: How were the hyperparameters for MTS3 tuned? How were the baselines tuned?**
>
> All hyperparameters for MTS3 and baselines including Transformers were tuned using grid search. For all recurrent models (MTS3, HiP-RSSM, RKN, LSTM, and GRU), we use a similar encoder-decoder architecture across datasets to ensure a fair comparison but allow the latent state dimensions to vary based on hyperparameter search. Small variations from these encoder-decoder architecture hyperparameters can still lead to similar prediction performance as reported in the paper.
>
> **Question: Transformers are achieving impressive results in control as of late. Where might they be integrated into this work in the future, to improve scalability, generalization, etc.?**
>
> We do not see a way as of now to integrate transformers into the "exact inference" scheme for MTS3. However, a variational formulation of MTS3 (on the same graphical model that we propose) would allow for integrating transformers, especially to replace the set encoders used for Bayesian aggregation with transformer-style (self-attention-based) task encoders. Here the task posterior would be parameterized with transformers operating on sets of observations. Deriving an ELBO for variational approximations on the MTS3 graphical model is straightforward and would result in mathematically convenient decomposable objectives and can be a future research direction.
>
> **Weakness: Confusing figure placement and formatting issues**
>
> Thank you for the suggestions. These would be duly addressed in the final version.

---

> > ### Comment · Reviewer_tb9F · 2023-08-16
> >
> > I thank the authors for adequately addressing my concerns. I maintain my prior opinion of the paper.

---

> > > ### Author Response · Authors · 2023-08-19
> > >
> > > Thank you for your reply and positive outlook!!

---

### Author Rebuttal · Authors · 2023-08-09

We thank all the reviewers for their valuable suggestions / insightful questions and comments. We would like to post answers here to some common questions/weaknesses raised by multiple reviewers.

**1. Questions on whether the strong experimental results are out of learning at multiple time scales or some training strategies like imputation**

We hope the extensive ablations that we report in the **rebuttal pdf**, with different values of the discretization step (which changes the magnitude of time scale) show that it's in fact learning temporally abstract hierarchical world models that result in improved performance. We also compare with a "Flat"  single time scale MTS3. Note that all ablations/baselines were, trained with the best possible imputation schemes to ensure a fair comparison. Moreover, we provide intuitive prediction plots that explain to some degree how MTS3 behaves when the discretization step $H$ is changed.

**2. A general definition and implementation sketch for MTS3 with arbitrary hierarchies. How to choose H.**

**Definition:** An $N$-level MTS3 is defined as a family of $N$-state space models, $\{S_0, S_1, ..., S_{N-1}\}$. Each of the state space models $S_i$ is given by $S_i = (Z_i, A_i, O_i, f_i, h_i, H_i \Delta t, L_i)$, where $Z_i$ is the state space, $A_i$ the action space, and $O_i$ the observation space of the SSM. The parameter $H_i \Delta t$ denotes the discretization time-step and $f_i$ and $h_i$ the dynamics and observation models, respectively. Here, $l_i \in L_i$ is a task descriptor that parametrizes the dynamics model of the SSM and is held constant for a local window of $H_{i+1}$ steps. $l_i$ is a function of the latent state of SSM one level above it, i.e., $S_{i+1}$. The boundary cases can be defined as follows: for $i=0$, $H_0 = 1$. Similarly, for $i=N-1$, the latent task descriptor $L_i$ is an empty set. For all $i$, $H_i < H_{i+1}$.

**Choosing Discretization Step:** Though we recommend searching for $H_i$ as a hyperparameter, as a general rule of thumb, it can be chosen as $H_i = (\sqrt[N]{T})^i$, where $T$ is the maximum prediction horizon required / episode length. This ensures that very long recurrences are divided between smaller equal-length task-reconfigurable local SSM windows (of length $\sqrt[N]{T}$) spread across several hierarchies.

**Computation and Implementation:** From a computational standpoint, the higher-level temporal abstractions are inferred by the aggregation of $H_i$ observations/actions at each level $i$. However, since we derive this aggregation as a permutation-invariant set operation, this can be efficiently parallelized. The aggregation rules derived can be thought of as simple "probabilistic attention," where attention weights are given by the learned variances in the set encoder and has a linear computational complexity of $O(H_i)$, when similar deep-set operations (self-attention) in transformers have a complexity of $O(H_i^2)$.

**3. The advantage over transformers / Can transformers be integrated into MTS3 formalism?**

The goal of this paper is to come up with a formal probabilistic framework to learn world models leveraging hierarchical temporal abstractions. This also lays a foundation for designing more principled hierarchical planning methods, that can leverage control as an inference framework. Though we outperform Transformers on several benchmarks, we think Transformers (or any stateful RNN) that operate at multiple timescales taking inspiration from this formalism can be a promising alternative research direction as pointed out in line 260 in the main paper. But coming up with such an architecture in a non-ad-hoc way is not trivial in our opinion.

There are also debates within the community as to whether LLMs/ Transformers capture causal relationships [1] and can autoregressive models like GPTs be effective for planning and reasoning [2].

We think our model combines the benefit of both worlds, set-based processing of temporal information as in Transformers via principled
aggregation schemes (a sort of probabilistic attention), yet maintaining the sequential dependency of RNNs which enforces causality by design. They also use a fraction of parameters as Transformer baselines and are more computationally efficient during inference as discussed in the previous section.

**Additional Points**

We are attaching a [zip file](https://drive.google.com/file/d/103fbLz1ahrfbSCGN0Ko58A7v76EVLPlu/) to our unofficial code. We once again thank all the reviewers for their insightful comments and questions. We hope we were able to answer most of the questions raised by the reviewers. Please let us know if you need more clarification.

**Reference**

[1] https://www.cclear.cc/2023/CLeaR23_roundtable_discussion.pdf

[2] LeCun, Yann. "A path towards autonomous machine intelligence version 0.9. 2, 2022-06-27." Open Review 62 (2022).

[3] Nagabandi et al. "Learning to adapt in dynamic, real-world environments through meta-reinforcement learning." ICLR (2018).

[4] Shaj  et al. "Hidden parameter recurrent state space models for changing dynamics scenarios." ICLR (2022).

[5] Dayan and Hinton. "Feudal reinforcement learning." NIPS (1992).

[6] Becker et al. "Recurrent Kalman networks: Factorized inference in high-dimensional deep feature spaces." ICML, 2019.

[7] Shaj et al. "Action-conditional recurrent Kalman networks for forward and inverse dynamics learning." CoRL, 2021.

[8] Wilson and Izmailov. "Bayesian deep learning and a probabilistic perspective of generalization." Neurips (2020):

[9] Volpp, Michael, et al. "Bayesian context aggregation for neural processes." ICLR. 2020.

[10] Singh, Gautam, et al. "Sequential neural processes." Neurips (2019).

[11] Lusch et al. "Deep learning for universal linear embeddings of nonlinear dynamics." Nature Communications (2018): 4950.

[12] Weissenbacher et al. "Koopman q-learning: Offline reinforcement learning via symmetries of dynamics." ICML, 2022.

---

### Decision · Program_Chairs · 2023-09-21

**Decision:**

Accept (spotlight)

**Comment:**

The paper presents a novel probabilistic formalism called Multi Time Scale State Space (MTS3) to allow intelligent agents to learn world models that operate at multiple levels of temporal abstraction. Through a computationally efficient inference scheme, the proposed model achieves highly accurate long-horizon predictions and uncertainty estimates in both simulated and real-world dynamical systems.

Reviewers have highlighted the paper's strengths in offering a novel closed-form multi-time scale inference approach and robust performance on several benchmarks. The formal derivation of the method and comparison against a wide range of valid baselines were also commended, as were the experiments that span both simulators and robots. However, the paper could further benefit from addressing limitations such as the two-level temporal abstraction and the model's exclusive focus on predictions. Some reviewers also pointed out the potential for a more general framework and noted some weaknesses in the baselines used for comparison. Despite these areas for improvement, some of which can be addressed in revisions, the paper stands as a high-quality contribution to the field and is strongly recommended for acceptance.